# DISSECTING ARBITRARY-SCALE SUPER-RESOLUTION CAPABILITY FROM PRE-TRAINED DIFFUSION GENERATIVE MODELS

## ABSTRACT

Diffusion-based Generative Models (DGMs) have achieved unparalleled performance in synthesizing high-quality visual content, opening up the opportunity to improve image super-resolution (SR) tasks. Recent solutions for these tasks often train architecture-specific DGMs from scratch, or require iterative fine-tuning and distillation on pre-trained DGMs, both of which take considerable time and hardware investments. More seriously, since the DGMs are established with a discrete pre-defined upsampling scale, they cannot well match the requirements of enhancing arbitrary images to a target resolution like 1080p, 2K, where a unified model adapts to arbitrary upsampling scales from the low-resolution to the target resolution, instead of preparing a series of distinct models for each case, we name it as TRSR. These limitations beg an intriguing question: *can we identify the TRSR capability of existing pre-trained DGMs without the need for distillation or fine-tuning?* In this paper, we take a step towards resolving this matter by proposing Diff-SR, a TRSR attempt based solely on pre-trained DGMs, without additional training efforts. It is motivated by an exciting finding that a simple methodology, which first injects a specific amount of noise into the low-resolution images before invoking a DGM's backward diffusion process, outperforms current leading solutions. The key insight is determining a suitable amount of noise to inject, *i.e.,* small amounts lead to poor low-level fidelity, while over-large amounts degrade the high-level signature. Through a finely-grained theoretical analysis, we propose the *Perceptual Recoverable Field* (PRF), a metric that achieves the optimal trade-off between these two factors. Extensive experiments verify the effectiveness, flexibility, and adaptability of Diff-SR, demonstrating superior performance to state-of-the-art solutions under diverse TRSR environments.

## 1 INTRODUCTION

Over the last decade, the deep neural models (*e.g.,* EDSR (Lim et al., 2017), ESRGAN (Wang et al., 2018)) have significantly promoted the development of super-resolution (SR) techniques. However, it is still challenging for them to match the requirements of target-resolution super-resolution (TRSR) tasks (Hui et al., 2019). The primary target of TRSR is to provide a unified model for arbitrary upsampling scales from the low-resolution to the target resolution, instead of training a series of distinct models for each case. This requirement is practical as individuals tend to prefer images and videos in specific resolutions such as 1080p or 2K. However, it is common for the images they receive to be in varying resolutions. Traditional SR models are often customized to a specific integer scale setting (*e.g.,* $2\times$). Thus, when dealing with a larger scale (*e.g.,* $4\times$) during inference, a natural way is to cascade a $2\times$ SR model twice which often suffers from losing high-fidelity details (Yang et al., 2014) or train a new $4\times$ SR model from scratch (Yang et al., 2018). Besides, due to the fixed network architecture, it is hard to adapt the model to non-integer scale SR tasks, *e.g.,* $2.7\times$. Observing these issues, the recent LIIF (Chen et al., 2021) and its variations (Wang et al., 2021; Xu et al., 2021) try to learn a continuous function representing high-resolution images. However, their upscaled images still suffer from unacceptable structural distortion and fidelity loss.

Recently, Diffusion-based Generative Models (DGMs) (Sohl-Dickstein et al., 2015), have achieved remarkable success in synthesizing high-quality visual content (Rombach et al., 2022; Ramesh

et al., 2022; Dhariwal & Nichol, 2021b; Croitoru et al., 2022; Harvey et al., 2022; Kawar et al., 2022a; Song et al., 2021b). Due to this unique strength (Dhariwal & Nichol, 2021a), DGMs have opened up the opportunity to handle image SR tasks (Saharia et al., 2021). Although DGMs can be implemented with large freedom, in this paper, we will exclusively demonstrate our methodology based on Denoising Diffusion Probabilistic Modeling (DDPM) (Ho et al., 2020), which is a pertinent case belonging to the DGM family. Generally, DDPM is inspired by non-equilibrium thermodynamics (Sohl-Dickstein et al., 2015). It defines a Markov chain (Ching & Ng, 2006) of diffusion steps by gradually adding Gaussian noise into data, and learns to reverse the diffusion process to reconstruct data samples from the noise (Weng, 2021). For image synthesis, DGMs start from a generative seed based on Gaussian noise, and then iteratively denoise it to obtain a clear image with high perceptual quality. Despite the benefit, most recent solutions often train architecture-specific DGMs from scratch, or require the efforts of fine-tuning (Hu et al., 2021; Gal et al., 2022; Roich et al., 2022) and distillation (Salimans & Ho, 2022; Song et al., 2023; Luhman & Luhman, 2021), both of which take considerable time and hardware investments. More seriously, the DGMs are established based on a discrete pre-defined upsampling scale, limiting their TRSR performance when adapting to a different scale during inference.

These limitations raise an interesting question – *can we identify the TRSR capability from an existing pre-trained DGM without additional efforts of fine-tuning or distillation?* Surprisingly, we find a simple but effective methodology to resolve this matter, *i.e.,* injecting a specific amount of noise into the low-resolution (LR) image before invoking a DGM's backward diffusion process. Our motivation is that since the DGMs can generate high-quality visual content in a fixed resolution, we can utilize this property to control a DGM's pipeline and adapt it to the TRSR environments. The key here is to determine a suitable amount of noise to inject. We develop insights that the injected noise affects the TRSR capability in both low-level fidelity measure and high-level signature of generated content. From this perspective, we prove the feasibility of this methodology in theory and deduce the *Perceptual Recoverable Field* (PRF). This key concept indicates how much noise could be injected to guarantee a good recovery quality for different upsampling scales. Based on these theoretical fundamentals, we deeply analyze the rationale of DGM's image TRSR capacity and give a mathematical analysis of the suitable noise injection level to obtain the desired recovery quality.

We implement our methodology as Diff-SR, a TRSR attempt based on a single pre-trained DGMs solely. Diff-SR just involves the inference process of DGMs. Diff-SR injects a specific amount of noise into the LR images and provides a unified generative starting point for visual details recovery. By invoking the reverse diffusion process, Diff-SR can restore the noisy LR images into the high-resolution version, with similar perceptual quality as the ground truth. Previous work, such as SDEDit(Meng et al., 2022), has demonstrated the potential of utilizing noise injection for image editing purposes. Building upon this foundation, our study aims to further showcase its effectiveness in conducting TRSR. Additionally, we provide a comprehensive analysis and evidence to substantiate our assertion, distinguishing our approach from SDEdit. The contributions of our work are as follows:

- **Efficient Methodology.** We take an attempt to identify a DGM's TRSR capability without distillation or fine-tuning. Excitingly, we find a simple but effective methodology for this matter that injects a specific amount of noise into the low-resolution image before invoking a DGM's backward diffusion process.

- **Theoretical Guarantee.** We establish theoretical analysis to support our methodology and quantify the TRSR capacity by deducing the key concept called *Perceptual Recoverable Field* (PRF). Based on these fundamentals, we provide mathematical analysis to guarantee the noise injection strength for handling different TRSR tasks.

- **Competitive Performance.** We implement our methodology as Diff-SR, a novel TRSR solution based solely on a single pre-trained DGM. Evaluations based on several datasets verify the competitiveness of Diff-SR over the current leading solutions.

## 2 BACKGROUND

### 2.1 FORWARD DIFFUSION PROCESS

For diffusion model, it models the whole forward process as a Markov chain. We add a small amount of Gaussian noise for each step in the chain to convert the original image to a low-quality version.

Each step is modeled by a Gaussian distribution $q(\mathbf{x}_t|\mathbf{x}_{t-1}) = \mathcal{N}(\mathbf{x}_t; \sqrt{1-\beta_t}\mathbf{x}_{t-1}, \beta_t\mathbf{I})$ where the noise strength is controlled by a variance schedule $\{\beta_t \in (0,1)\}_{t=1}^T$. By utilizing the property of Gaussian distribution, we can formulate the distribution of $x_t$ given $x_0$ as:

$$q(\mathbf{x}_t|\mathbf{x}_0) = \mathcal{N}(\mathbf{x}_t; \sqrt{\bar{\alpha}_t}\mathbf{x}_0, (1-\bar{\alpha}_t)\mathbf{I}), \tag{1}$$

where $\alpha_t = 1 - \beta_t$ and $\bar{\alpha}_t = \prod_{i=1}^t \alpha_i$. Usually, the noise strength increases along with time $\beta_1 < \beta_2 < ... < \beta_T$. Therefore, we have $\bar{\alpha}_1 > \bar{\alpha}_2 > ... > \bar{\alpha}_T$. With this condition distribution, we can derive the posterior distribution of $x_{t-1}$ conditioned by $x_t, x_0$ as:

$$q(\mathbf{x}_{t-1}|\mathbf{x}_t, \mathbf{x}_0) = \mathcal{N}(\mathbf{x}_{t-1}; \tilde{\boldsymbol{\mu}}(\mathbf{x}_t, \mathbf{x}_0), \tilde{\beta}_t\mathbf{I}). \tag{2}$$

## 2.2 BACKWARD DIFFUSION PROCESS

The target of a DGM is to maximize the likelihood probability $p_\theta(x_0)$. Based on the negative log-likelihood theorem, we can maximize $p_\theta(x_0)$ and optimize the variational upper bound by introducing a KL-divergence term (Hershey & Olsen, 2007; Joyce, 2011). According to the preliminary formulation mentioned by Eq. (1) and Eq. (2), we can deduce the variational lower bound $L_{VLB}$ as:

$$-\log p_\theta(\mathbf{x}_0) \leq -\log p_\theta(\mathbf{x}_0) + D_{\text{KL}}(q(\mathbf{x}_{1:T}|\mathbf{x}_0)\|p_\theta(\mathbf{x}_{1:T}|\mathbf{x}_0))$$
$$= \underbrace{L_T}_{\text{Forward error}} + \underbrace{L_{T-1} + \cdots + L_0}_{\text{Backward error}} \tag{3}$$

where:

$$L_T = D_{\text{KL}}(q(\mathbf{x}_T|\mathbf{x}_0) \| p_\theta(\mathbf{x}_T)) = C_T, L_0 = -\log p_\theta(\mathbf{x}_0|\mathbf{x}_1).$$
$$L_t = \mathbb{E}_{\mathbf{x}_0, \boldsymbol{\epsilon}}\left[\frac{(1-\alpha_t)^2}{2\alpha_t(1-\bar{\alpha}_t)\|\boldsymbol{\Sigma}_t\|_2^2}\|\boldsymbol{\epsilon}_t - \boldsymbol{\epsilon}_\theta(\mathbf{x}_t, t)\|^2\right] \tag{4}$$

Note that $L_T$ reflects the forward error since it models the difference between the forward distribution $q(\mathbf{x}_T|\mathbf{x}_0)$ and the distribution of the neural network output. The other parts are the backward error, which measures the difference between the true backward distribution $q(\mathbf{x}_{t-1}|\mathbf{x}_t, \mathbf{x}_0)$.

## 3 METHOD

### 3.1 OBSERVATIONS FROM DGM DENOISING PROCESS

As illustrated in Figure. 1, when initiating the backward process at timestep $t$ rather than $T$, a distinct output can still be obtained from the noisy input image. In this context, we denote $\bar{L}_t$ as the sampling error when commencing at $\mathbf{x}_t$, where $\mathbf{x}_t$ represents the outcome after injecting $t$ forward noise steps to the original image. Based on this, the following lemma is established:

**Lemma 1.** *The error between the output image and the GT* $\mathbf{x}_0$ *can be formulated as:*

$$\bar{L}_t = C_t + \sum_{i=1}^{t-1}\left[\frac{(1-\alpha_i)^2}{2\alpha_i(1-\bar{\alpha}_i)\|\boldsymbol{\Sigma}_i\|_2^2}E_i\right] + L_0, \tag{5}$$

where $E_i = \|\boldsymbol{\epsilon}_i - \boldsymbol{\epsilon}_\theta(\sqrt{\bar{\alpha}_i}\mathbf{x}_0 + \sqrt{1-\bar{\alpha}_i}\boldsymbol{\epsilon}_i, i)\|^2$. Note that $\bar{L}_t$ models the error between the output image and the original image. $E_i$ is the training loss of the neural network and can be estimated empirically once the network converges. $E_i$ is actually the loss function of the diffusion model. When the neural network converges, this value can be regarded as a constant $E_0$. $\boldsymbol{\Sigma}_i$ is the variance of the reverse process which changes along with time $i$. As there are many research about how to speed up the sampling process of difussion model like DDIM (Song et al., 2021a), this parameter may differ in the different sampler. For the basic DDPM (Ho et al., 2020), $\boldsymbol{\Sigma}_t^{DDPM} = \tilde{\beta}_i\mathbf{I}$, for DDIM, $\boldsymbol{\Sigma}_t^{DDIM} = \frac{1-\bar{\alpha}_{i-1}}{1-\bar{\alpha}_i} \cdot \frac{1-\bar{\alpha}_i}{\bar{\alpha}_{i-1}}\mathbf{I}$. Detailed proof can be found in the supplementary material of §B.

Our first key observation is that this error can reflect the reversibility of the noisy image back to its original image. As depicted in Figure. 1, the behavior of $\bar{L}_t$ manifests a pronounced surge initially, where $t/T < 0.4$. During this phase, we note that the reconstructed image undergoes negligible

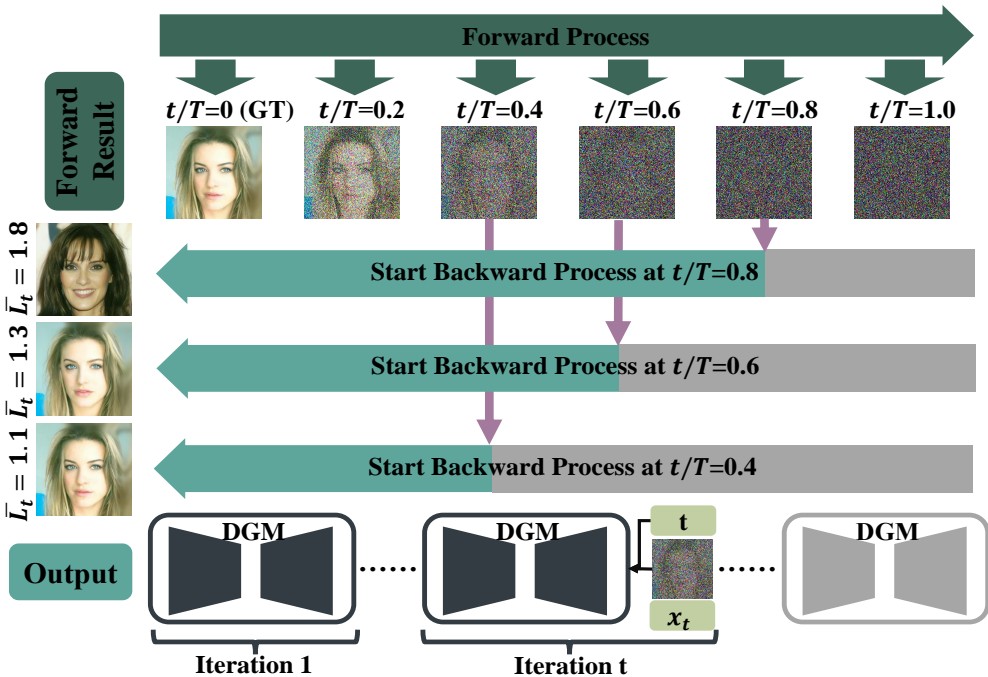

Figure 1: Illustration of DGM denoising process.

changes in comparison to the original image. Subsequently, a gradual and smoother transition occurs up to around $t/T = 0.6$. Throughout this interval, the reconstructed image largely retains the essential features of the original version, albeit with slight deviations in certain details. In the final phase, the reconstructed image diverges significantly from the original rendition. This intriguing phenomenon serves as a catalyst for further exploration, prompting us to investigate the potential applications of this characteristic.

## 3.2 POTENTIAL OF DISSECTING ASSR CAPABILITY

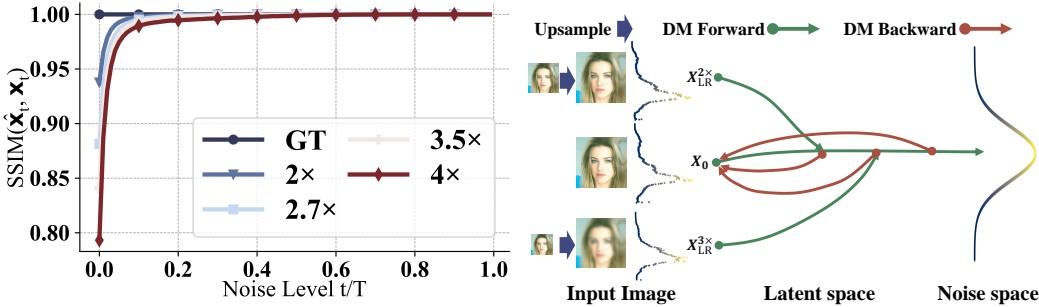

(a) Similarity after injecting noise into LR images.    (b) Probability flow of DM process with different scale.

Figure 2: By injecting a specific amount of noise into the LR images, it is possible to restore the LR images to the HR version. However, the final perceptual quality of the recovered HR images may change with the injected noise level, which is controlled by noise level $t/T$.

Then, inspired by our preliminary experiment, we found that after injecting some noise to low-resolution images $\hat{\mathbf{x}}$, the sampling results $\hat{\mathbf{x}}_t$ show high similarity with the high-resolution images $\hat{\mathbf{x}}$ as shown in Figure. 2a. This adding noise process is the sampling process from the forward distribution $\mathbf{x}_t \sim q(\mathbf{x}_t|\mathbf{x}_0)$. The high similarity is because their distributions become similar. The distribution similarity can be measured by KL-divergence $D_{\mathrm{KL}}(q(\mathbf{x}_t|\mathbf{x}_0) \parallel q(\hat{\mathbf{x}}_t|\hat{\mathbf{x}}_0))$ where we have deeper analysis in § 3.3. This inspires us to explore whether we can recover the high-resolution image with these noised low-resolution images, so we try to use different resolutions to conduct reserves steps after injecting a certain amount of noise into them. The source resolution is $256 \times 256$,

we convert the source image to different versions, and the result is demonstrated in the Figure. 3. According to the experiment, we can easily recover the original image when the resolution is near the original resolution (*i.e.,* downsampling the original image to $2.6\times$ scale), just inject $20\%$ noise into the degraded image, then we can recover the high-resolution image. However, as the resolution decreases, we should inject more noise, as shown in the $4.5\times$ scale downsampling, when we add $20\%$ noise and start the reverse process at this point, it ends up returning a blurry picture just as the input data, So we need to add more noise to the input data, then it can return a more clear picture that maintains the majority of the source image.

Actually, by injecting noise into the LR images, we make the probability flow of LR images overlap with the HR images as illustrated in Figure. 2b. As the forward flow always ends at the Gaussian noise space, we can identify a uniform latent space that can be reached by both HR and LR images through noise injection. Moreover, Pretrain DGM is trained for reversing the forward flow of HR images, so we can leverage the model to enhance the latent feature of LR images and align them to follow the backward flow of HR images, ultimately generating HR outputs. However, there is a tradeoff which is if we add too much noise, it will finally return totally different images.

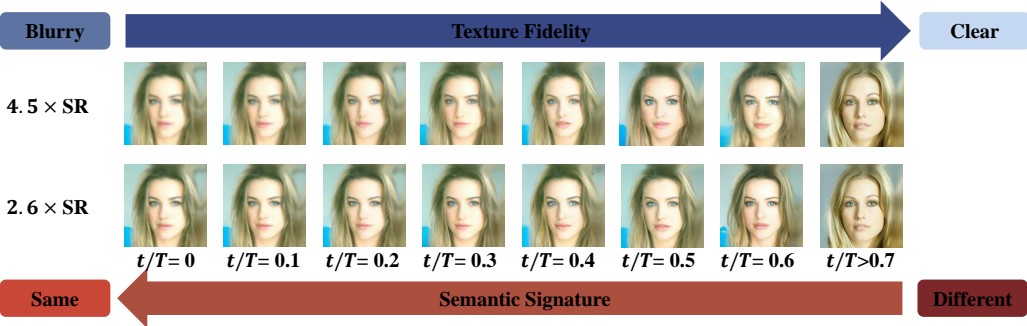

Figure 3: Image SR results of a pre-train DGM with different steps of noise injection. Take both $4.5\times$ and $2.6\times$ upsampling scales as the example, we inject different $t$ steps of noise into the input LR image, and then conduct the DGM's reverse process from this step to generate the HR version. We can see that the step numbers impact both texture fidelity and semantic signature, which are with opposite proportional relations to step $t$.

### 3.3 ANALYSIS OF RECOVERY ERROR

In this section, we analyze how to control the amount of noise injected and explain why it works. Note that $p_\theta$ is a well pretrained Diffusion model. By just injecting some noise into the low-resolution images and changing the start point of the reverse diffusion process, we can now use $p_\theta$ to conduct super-resolution tasks.

With a little abuse of symbols, we use $\mathbf{x}_t, \hat{\mathbf{x}}_t$ to represent the result after we inject $t$ steps noise to the original high-resolution images and low-resolution images, respectively. $\bar{L}_t$ is the upper error bound between the ground truth image $\mathbf{x}_0$ and the generated image when we use $\mathbf{x}_0$ as the model input. When we use $\hat{\mathbf{x}}_0$ as the model input and want to get a clear output image, the backward process is almost the same as the backward process when we use $\mathbf{x}_0$ as input because it mainly depends on the neural network parameter. The main difference comes from the forward error. Here, we use $\mathcal{L}_t$ to represent the error between the ground truth image $\mathbf{x}_0$ and the generated image $\tilde{\mathbf{x}}$.

**Definition 1.** *In the inference period, given the original high-resolution image $\mathbf{x}$ (i.e., the ground truth) and the compressed low-resolution version $\hat{\mathbf{x}}$, we first inject $t$ steps of Gaussian noise into $\hat{\mathbf{x}}$ to obtain the noisy version $\hat{\mathbf{x}}_t$, then feed $\hat{\mathbf{x}}_t$ into DGM as the generative seed, and finally, reverse the diffusion process also through $t$ denoising steps to generate the recovered image $\bar{\mathbf{x}}$. Therefore, the entire recovery error $\mathcal{L}_t$ between the recovered image $\bar{\mathbf{x}}$ and ground-truth $\mathbf{x}$ can be formulated as:*

$$\mathcal{L}_t = D_{\mathrm{KL}}(q(\mathbf{x}_t|\hat{\mathbf{x}}_0) \parallel p_\theta(\mathbf{x}_t)) + \sum_{i=0}^{t-1} L_i. \tag{6}$$

Based on the definition of the recovery error, our next step is to analyze which are the key terms impacting this error.

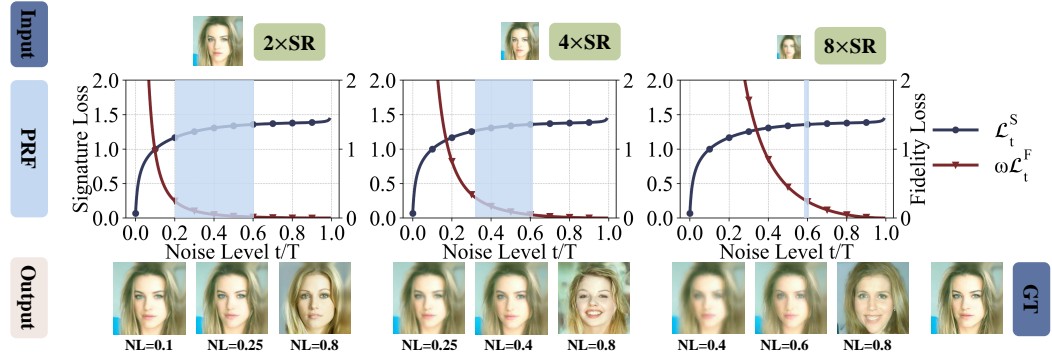

Figure 4: The visualization of PRF under different upsampling scales. The PRF satisfies the constraints of both signature and fidelity loss, indicating a suitable amount of noise injection.

**Theorem 1.** *The recovery error $\mathcal{L}_t$ can be resolved as two terms: the signature loss $\mathcal{L}_t^S$ and fidelity loss $\mathcal{L}_t^F$, where the former reflects the structural similarity of the entire visual content while the latter measures the smoothness of detailed textures. The formulation can be described as:*

$$\mathcal{L}_t \triangleq \bar{L}_t + \left[ K_t \parallel \mathbf{x} - \hat{\mathbf{x}}_0 \parallel^2 + A_t \right]$$
$$\triangleq \underbrace{\mathcal{L}_t^S}_{\text{Signature Loss}} + \underbrace{\omega \mathcal{L}_t^F}_{L_1 \text{ Fidelity Loss}}, \tag{7}$$

where $\omega$ is a hyperparameter to guarantee these two loss terms are of the same magnitude. Empirically, we set $\omega = 0.004$. Note that $K_t, A_t$ are two intermediate variables decreasing with $t$. Their detailed descriptions and the complete proof can be found in the supplementary material of §C. This theorem reveals that the signature loss increases with $t$ while the fidelity loss decreases with $t$. A lower signature loss helps preserve visual similarity, *i.e.,* in Figure. 3, the upscaled image holds a similar human face as the ground truth. Besides, a lower fidelity loss serves as image deblurring, *i.e.,* in Figure. 3, the upscaled image provides a clear human face with sharp texture. By jointly optimizing these two terms, we can finally upscale the images with high perceptual quality.

## 3.4 DETERMINING NOISE INJECTION VIA PERCEPTUAL RECOVERABLE FIELD

**Remark.** *To restore the low-resolution image to a high-resolution one with diffusion model, we should inject $t$ steps noise to the low-resolution image $\hat{\mathbf{x}}_0$ so that both $\mathcal{L}_t^S, \omega \mathcal{L}_t^F$ are less than a threshold, then with the capacity of diffusion model, it will restore the noisy blur image $\hat{\mathbf{x}}_t$ to a clear high-quality image $\tilde{\mathbf{x}}$. As $t$ controls the amount of injected noise, we call the range of $t$ satisfying the above constraints as **Perceptual Recoverable Field (PRF)**. Consequently, searching the PRF corresponds to determining a suitable amount of noise to inject. The searching process can be formulated as solving the following problem.*

$$\arg\min_t \quad \mathcal{L}_t$$
$$\text{s.t.} \quad \mathcal{L}_t^S \leq \mathcal{C}^S \tag{8}$$
$$\omega \mathcal{L}_t^F \leq \mathcal{C}^F, \tag{9}$$

where the first constraint in Eq. (8) ensures that the injected noise will not destroy the content of the input image and result in a wrong output image. Meanwhile, the second constraint in Eq. (9) ensures that the recovered image $\tilde{\mathbf{x}}$ will become clearer compared with the input low-resolution image $\hat{\mathbf{x}}_0$. $\mathcal{C}^S$ and $\mathcal{C}^F$ are two constant thresholds.

We illustrate this decision process in Figure. 4, and we use **NL** here to represent noise level. For different resolutions, the acceptable PRFs are different. For $2\times$ super-resolution, it just needs to inject about $20\%$ noise. Then starting the diffusion sampling process at this point, we can get a high-quality output. For those downsampling in a higher scale like $4\times$ upsampling, we need to inject about $40\%$ noise to get an acceptable output. For $8\times$ upsampling, it doesn't have a PRF area, which means it

is hard to restore the original image. Actually, we can still get a reasonable output when we add $50 - 65\%$ noise to this input. But, compared with other resolutions which have a wider PRF, this output seems to be more blurred and more different in some areas.

## 4 EXPERIMENTS

### 4.1 EXPERIMENTAL SETUP

In our experimental evaluation, we thoroughly investigate the influence of noise injection in different settings. Consistent with the state-of-the-art architecture of DGM, the pre-trained DGM utilizes the U-Net (Ronneberger, 2017) architecture as the backbone. For comprehensive information regarding implementation details and hyperparameters, we provide the supplementary material in §D. Our experiments are conducted on four publicly available datasets specifically designed for evaluating image editing tasks: DIV2K (Agustsson & Timofte, 2017), Set5 (Bevilacqua et al., 2012), Set14 (Zeyde et al., 2010), and Urban100 (Huang et al., 2015). Prior to conducting the experiments, we apply preliminary processing steps, including center cropping and resizing the images to a standardized size of $256 \times 256$. It is important to note that the pre-trained DGM employed in our Diff-SR method is solely trained on images of the $256 \times 256$ resolution setting and does not have access to any downsampled images. Following the methodology outlined in (Rombach et al., 2022), we assess the quality of image editing using both perceptual-based metrics, such as Fréchet Inception Distance (FID) (Heusel et al., 2017), which aligns closely with human perception, as well as distortion-based metrics like Peak signal-to-noise ratio (PSNR) (Horé & Ziou, 2010) and structural similarity index measure (SSIM) (Wang et al., 2004). By employing these metrics, we can comprehensively evaluate the impact of noise injection on both fidelity and perception fields (Lee et al., 2022). To establish a fair comparison, we benchmark our method against several baseline methods provided by OpenMMLab (OpenMMLab, 2022), latest DM-based method including DDRM(Kawar et al., 2022b), DDNM(Wang et al., 2023), LDM-SR3(Rombach et al., 2022) and other methods for like LTE(Lee & Jin, 2022), ITSRN(Yang et al., 2021), SwinIR(Liang et al., 2021). We download their pre-trained models from their official GitHub repositories to ensure fairness in the evaluation.

### 4.2 PERFORMANCE COMPARISON

Table 1: Comparison with more baseline solutions, where the **red** and blue colors indicate the best and the second-best performance, respectively.

| Model | Scale | #Params | #Mult-Adds | DIV2K | | | Set5 | | | Set14 | | | Urban100 | | |
|---|---|---|---|---|---|---|---|---|---|---|---|---|---|---|---|
| | | | | FID↓ | PSNR↑ | SSIM↑ | FID↓ | PSNR↑ | SSIM↑ | FID↓ | PSNR↑ | SSIM↑ | FID↓ | PSNR↑ | SSIM↑ |
| Bicubic | 4× | NA | NA | 14.963 | 30.451 | 0.679 | 7.134 | 32.214 | 0.808 | 14.553 | 30.956 | 0.707 | 16.468 | 30.171 | 0.624 |
| DDNM | 4× | 552M | 3.31G | 5.226 | 31.377 | 0.781 | 2.363 | 33.716 | 0.866 | 5.527 | 31.888 | 0.777 | 6.038 | 31.135 | 0.764 |
| DDRM | 4× | 552M | 3.31G | 7.752 | 30.478 | 0.706 | 5.217 | 32.108 | 0.836 | 9.091 | 30.774 | 0.710 | 8.005 | 30.771 | 0.735 |
| LDM-SR3 | 4× | 150M | 5.3G | 2.805 | 29.361 | 0.512 | 13.140 | 29.663 | 0.517 | 2.394 | 29.498 | 0.528 | 2.209 | 29.695 | 0.589 |
| EDSR | 4× | 1.51M | 8.13G | 7.113 | 31.538 | 0.808 | 3.894 | 34.019 | 0.907 | 7.865 | 32.068 | 0.801 | 7.049 | 31.274 | 0.793 |
| ESRGAN | 4× | 16.72M | 73.48G | 6.766 | 31.622 | 0.819 | 3.849 | 34.067 | 0.912 | 7.673 | 32.124 | 0.807 | 5.672 | 31.408 | 0.815 |
| LTE | 4× | 11.95M | 149.86G | 6.530 | 31.658 | 0.823 | 3.607 | 34.073 | 0.915 | 7.354 | 32.158 | 0.811 | 5.295 | 31.471 | 0.826 |
| ITSRN | 4× | 22.6M | 12.05G | 7.356 | 31.513 | 0.806 | 3.924 | 34.011 | 0.907 | 7.828 | 32.055 | 0.802 | 7.835 | 31.232 | 0.786 |
| LIIF | 4× | 1.57M | 5G | 7.111 | 31.548 | 0.809 | 3.779 | 34.003 | 0.910 | 7.866 | 32.073 | 0.802 | 6.879 | 31.279 | 0.794 |
| SwinIR | 4× | 11.75M | 9.46G | 6.402 | 31.671 | 0.824 | 3.625 | 34.090 | 0.916 | 7.354 | 32.153 | 0.811 | 5.059 | 31.478 | 0.824 |
| **Diff-SR(Ours)** | 4× | 35.71M | 40.56G | 1.088 | 32.450 | 0.924 | 0.196 | 34.732 | 0.961 | 0.281 | 34.982 | 0.964 | 0.915 | 32.148 | 0.910 |
| Bicubic | 2× | NA | NA | 4.879 | 32.434 | 0.889 | 3.417 | 35.044 | 0.929 | 5.134 | 33.064 | 0.879 | 6.129 | 31.758 | 0.848 |
| DDNM | 2× | 552M | 3.31G | 0.599 | 33.491 | 0.898 | 0.305 | 34.669 | 0.876 | 0.402 | 33.663 | 0.876 | 0.499 | 32.982 | 0.887 |
| DDRM | 2× | 552M | 3.31G | 2.306 | 32.168 | 0.877 | 2.134 | 34.003 | 0.910 | 3.094 | 32.348 | 0.847 | 2.062 | 32.718 | 0.897 |
| LDM-SR3 | 2× | NA | NA | NA | NA | NA | NA | NA | NA | NA | NA | NA | NA | NA | NA |
| EDSR | 2× | 1.37M | 7.93G | 0.539 | 35.885 | 0.966 | 0.924 | 37.849 | 0.973 | 0.751 | 35.928 | 0.944 | 0.717 | 34.970 | 0.954 |
| ESRGAN | 2× | NA | NA | NA | NA | NA | NA | NA | NA | NA | NA | NA | NA | NA | NA |
| LTE | 2× | 11.95M | 149.86G | 0.489 | 36.300 | 0.970 | 0.907 | 37.899 | 0.974 | 0.681 | 36.279 | 0.952 | 0.588 | 35.492 | 0.962 |
| ITSRN | 2× | 22.6M | 12.05G | 0.620 | 35.758 | 0.965 | 0.989 | 37.707 | 0.971 | 0.810 | 35.889 | 0.943 | 0.881 | 34.786 | 0.952 |
| LIIF | 2× | 1.57M | 5G | 1.366 | 34.851 | 0.957 | 1.598 | 37.078 | 0.965 | 1.648 | 35.166 | 0.935 | 1.833 | 33.759 | 0.941 |
| SwinIR | 2× | 11.55M | 9.26G | 0.455 | 36.354 | 0.970 | 0.872 | 37.925 | 0.974 | 0.651 | 36.390 | 0.953 | 0.535 | 35.597 | 0.963 |
| **Diff-SR(Ours)** | 2× | 35.71M | 40.56G | 0.411 | 35.410 | 0.964 | 0.082 | 38.373 | 0.976 | 0.103 | 38.239 | 0.977 | 0.169 | 36.592 | 0.978 |

As demonstrated in Table 1, our model consistently achieves superior performance across the majority of datasets. At lower compression rates (e.g., 2×), while Diff-SR delivers commendable overall metrics, its advantage over certain baselines may not be substantial, as some alternatives manage to achieve similar performance levels. However, as the compression rate increases (e.g., 4×), the distinct superiority of Diff-SR becomes more evident. Higher compression rates inevitably result in the loss or degradation of original information, leading to visual artifacts, blurring, and other forms of distortion. Consequently, restoring images for other baselines becomes more challenging, even after retraining these models on the specific data. In contrast, Diff-SR maintains almost identical

performance, even under high-rate compression scenarios. Notably, Diff-SR exhibits more significant improvements in perception field metrics, such as FID, compared to fidelity field metrics like PSNR and SSIM. This discrepancy arises from the fact that PSNR and SSIM tend to favor mean squared error (MSE) regression-based techniques, which tend to be excessively conservative with high-frequency details. However, these metrics penalize synthetic high-frequency details that may not align well with the target image (Saharia et al., 2021). Consequently, diffusion-based models like SR3 and Diff-SR achieve higher FID scores while occasionally exhibiting lower PSNR and SSIM scores. The performance advantage of Diff-SR is further verified through visualizations, as illustrated in Figure. 5.

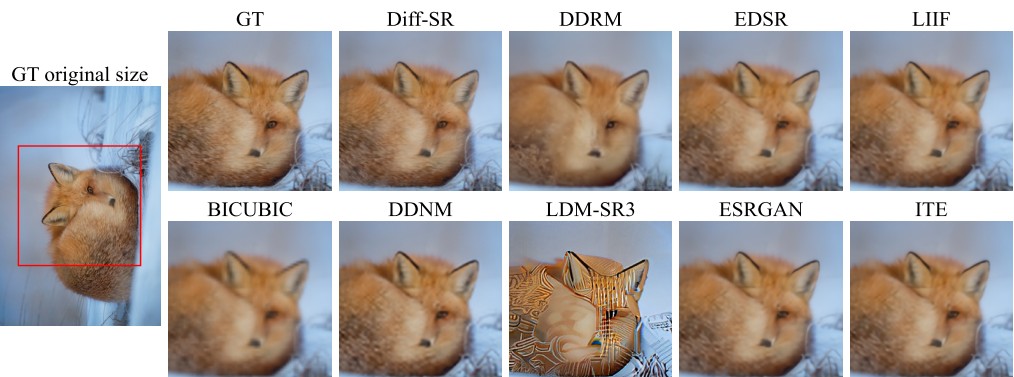

Figure 5: Performance visualization of different $4\times$ SR solutions, evaluated on DIV2K dataset. Additional results are provided in the supplementary material of §F. (**Zoom-in for best view**)

Importantly, when considering other baselines like EDSR, ESRGAN, achieving upsampling scales such as $2.7\times$ poses certain challenges. One approach involves employing a $2\times$ super-resolution model to enhance an image from $96 \times 96$ to $192 \times 192$ and subsequently using basic upsampling methods like Bicubic to upscale it to $256 \times 256$. Alternatively, another method involves using a $4\times$ super-resolution model to enhance the image and then downsampling the output to $256 \times 256$. However, both of these methods suffer from performance limitations due to the distortions introduced by further compression and the mismatch of compression scales. In contrast, our Diff-SR approach leverages a single pre-trained model to handle diverse scale super-resolution tasks. By simply adjusting the noise injection level, Diff-SR achieves excellent results for various scale SR scenarios. Notably, Diff-SR is capable of directly accepting image inputs of any scale (e.g., $2.7\times$, $3.5\times$) without the suffering of degradation caused by techniques like Bicubic interpolation.

Note that our primary objective is to explore the TRSR capacity of pretrain DGM so we didn't put much effort to minimize the amount of parameters or optimize the inference speed.

### 4.3 ABLATION STUDIES

#### 4.3.1 IMPACT OF UPSAMPLING SCALES

In the conducted ablation experiment, as illustrated in Figure. 6, we explored various super-resolution (SR) scales ranging from $1.6\times$ to $4.5\times$. The results revealed the necessity of adapting noise injection levels to suit different SR tasks. Initially, at the state $t/T = 0$, we observed a deterioration in metrics such as FID, PSNR, and SSIM as the resolution decreased. However, by employing Diff-SR to restore these images, their quality experienced significant improvement, particularly when the noise level was not excessively high. Figure. 6 presents the outcomes of the experiment. For a $2\times$ upscaling SR task, Diff-SR achieved FID (0.50), PSNR (34.2 dB), and SSIM (89.5) scores by employing a noise injection level of approximately 20%. Conversely, to attain similar performance in a $4\times$ upscaling SR task, Diff-SR required a noise injection level of approximately 40%, resulting in FID (0.55), PSNR (33.2 dB), and SSIM (88.4) scores. The observed discrepancies in these metrics are primarily attributable to the degradation of the input image. Nevertheless, when compared to the initial state, Diff-SR substantially mitigated these discrepancies. Importantly, this experiment emphasized the importance of avoiding excessive noise injection into the input images. As depicted in the figure, if the noise injection level exceeds 60%, all metrics exhibit a deterioration worse than that of the initial

state. This finding aligns with our preliminary observations, where excessively high noise injection levels led to visually clear output images but compromised their semantic content.

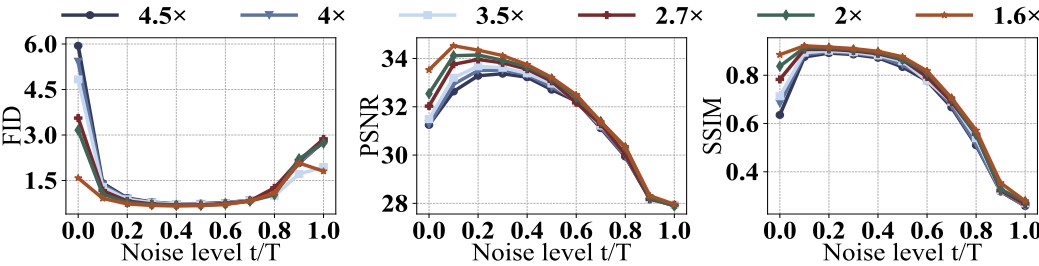

Figure 6: Ablation study of upsampling scales on REDS dataset.

### 4.3.2 IMPACT OF DGM PRE-TRAINING SCHEDULER

We further conducted an ablation study to investigate the impact of different hyperparameters on the required noise injection levels for achieving target-scale super-resolution. Notably, we identified the noise scheduler method as a significant influencing factor. The noise scheduler method regulates the strength of noise injection at each step within the DGM model, which is accomplished by manipulating the parameters $\alpha_t$ and $\beta_t$. To explore this, we examined several noise schedulers, including Cosine (Nichol & Dhariwal, 2021), Linear (Kingma et al., 2021), Scale Linear (Austin et al., 2021), Sigmoid (Chen, 2023), and Square Cosine (Rombach et al., 2022), which have been previously adopted in Stable Diffusion (Rombach et al., 2022). For this ablation experiment, we focused on a $4\times$ super-resolution scale. The results of this study, presented in Figure. 7, showcased the behavior of both methods starting from the same initial metric point and exhibited similar trends as the noise injection level varied. Among the various scheduler methods, the Square Cosine scheduler emerged as the most effective. It achieved a superior FID score (0.2), the second-best PSNR score (34.2 dB), and SSIM score (89%) within a noise injection level range of 10% to 70%. In contrast, schedulers such as Cosine and Linear, which were initially employed in the original DGM version (Ho et al., 2020), demonstrated poorer performance on these metrics at the same noise injection level. Additionally, the acceptable range of noise levels for satisfactory results was narrower compared to the Square Cosine scheduler.

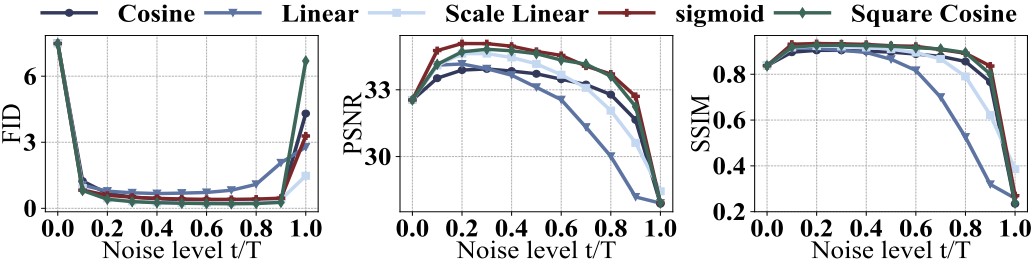

Figure 7: Ablation study of DGM pre-training scheduler on REDS dataset.

## 5 CONCLUSION

This research endeavor aims to explore the target-resolution super-resolution (TRSR) capabilities inherent in existing pre-trained diffusion-based generative models (DGMs), without necessitating additional fine-tuning or distillation efforts. We present Diff-SR, the TRSR approach that relies solely on pre-trained DGMs. The foundation of Diff-SR lies in a simple yet powerful observation: by introducing a specific level of noise into the low-resolution (LR) image prior to initiating the DGM's backward diffusion process, the desired recovery performance can be achieved. We further substantiate the feasibility of this methodology through theoretical analysis and introduce a fundamental metric known as the *Perceptual Recoverable Field* (PRF). The PRF metric quantifies the permissible amount of noise that can be injected to ensure high-quality recovery for various upsampling scales. Our extensive experiments demonstrate that Diff-SR surpasses existing state-of-the-art solutions across diverse super-resolution scenarios.

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

## A   NOTATIONS

Table 2: Notation list.

| Notation | Description |
|---|---|
| $\beta_t$ | The variance schedule of Diffusion model $\beta_t \in (0,1)$, where $t \in [1,T]$ |
| $\alpha_t$ | Defined as $\alpha_t = 1 - \beta_t$ |
| $\bar{\alpha}_t$ | Defined as $\bar{\alpha}_t = \prod_{i=1}^{t} \alpha_i$ |
| $\tilde{\beta}_t$ | The output based on $\bar{\alpha}_t, \beta_t, \tilde{\beta}_t = \frac{1-\bar{\alpha}_{t-1}}{1-\bar{\alpha}_t} \cdot \beta_t$ |
| $\mathbf{x}$ | The original high-resolution image, *i.e.,* the ground truth |
| $\mathbf{x}_0$ | The initial status before injecting Gaussian noise, *i.e.,* $\mathbf{x}_0 = \mathbf{x}$ |
| $\mathbf{x}_t$ | The final status after injecting Gaussian noise by $t$ steps to ground truth |
| $\hat{\mathbf{x}}$ | The low-resolution image compressed from $\mathbf{x}$ |
| $\bar{\mathbf{x}}$ | The recovered image based on $\hat{\mathbf{x}}$ |
| $t$ | The step number that injects Gaussian noise into $\hat{x}$, $t \in [0,T]$ |
| $\hat{\mathbf{x}}_0$ | The initial status before injecting Gaussian noise, *i.e.,* $\hat{\mathbf{x}}_0 = \hat{\mathbf{x}}$ |
| $\hat{\mathbf{x}}_t$ | The final status after injecting Gaussian noise by $t$ steps |
| $\tilde{\mu}_t$ | The mean of the reverse gaussian distribution |
| $\epsilon_t$ | The injected noise at timestep $t$ |
| $\epsilon_\theta$ | The predicted noise by neural network |
| $E_t$ | The prediction error between $\epsilon_t$ and $\epsilon_\theta$ |
| $\bar{L}_t$ | The accumulation error of $t$ steps denoising with $\mathbf{x}$ as input |
| $\mathcal{L}_t$ | The accumulation error of $t$ steps denoising with $\hat{\mathbf{x}}$ as input |
| $A_t, K_t$ | Derived intermediate variables decrease along with $t$ |
| $\mathcal{L}_t^S$ | The signature loss |
| $\mathcal{L}_t^F$ | The fidelity loss |
| $\omega$ | The hyperparameter that guarantees $\mathcal{L}_t^S$ and $\mathcal{L}_t^F$ with the same order of magnitude |

All the notations used in the supplementary material are listed in Table 2.

## B   PROOF OF **LEMMA 1.** IN SEC. 3.2

*Proof.* Following the idea of DDPM, when we start the reverse process at time step $t$ the variational upper bound to optimize the negative log-likelihood can be rewritten as:

$$
\begin{aligned}
-\log p_\theta(\mathbf{x}_0) &\leq -\log p_\theta(\mathbf{x}_0) + D_{\text{KL}}(q(\mathbf{x}_{1:t}|\mathbf{x}_0)\|p_\theta(\mathbf{x}_{1:t}|\mathbf{x}_0)) \\
&= \mathbb{E}_{q(\mathbf{x}_{0:t})}\left[\log \frac{q(\mathbf{x}_{1:t}|\mathbf{x}_0)}{p_\theta(\mathbf{x}_{0:t})}\right] \\
&= L_{VUB} \\
&= \bar{L}_t
\end{aligned}
\tag{10}
$$

Then follow the derivation of (Sohl-Dickstein et al., 2015), we have

$$
\begin{aligned}
\bar{L}_t &= \mathbb{E}_{q(\mathbf{x}_{0:t})}\Big[\log \frac{q(\mathbf{x}_{1:t}|\mathbf{x}_0)}{p_\theta(\mathbf{x}_{0:t})}\Big] \\
&= \mathbb{E}_q\Big[\log \frac{\prod_{i=1}^{t} q(\mathbf{x}_i|\mathbf{x}_{i-1})}{p_\theta(\mathbf{x}_t)\prod_{i=1}^{t} p_\theta(\mathbf{x}_{i-1}|\mathbf{x}_i)}\Big] \\
&= \mathbb{E}_q\Big[-\log p_\theta(\mathbf{x}_t) + \sum_{i=1}^{t} \log \frac{q(\mathbf{x}_i|\mathbf{x}_{i-1})}{p_\theta(\mathbf{x}_{i-1}|\mathbf{x}_i)}\Big] \\
&= \mathbb{E}_q\Big[-\log p_\theta(\mathbf{x}_t) + \sum_{i=2}^{t} \log \frac{q(\mathbf{x}_i|\mathbf{x}_{i-1})}{p_\theta(\mathbf{x}_{i-1}|\mathbf{x}_i)} + \log \frac{q(\mathbf{x}_1|\mathbf{x}_0)}{p_\theta(\mathbf{x}_0|\mathbf{x}_1)}\Big] \\
&= \mathbb{E}_q\Big[-\log p_\theta(\mathbf{x}_t) + \sum_{i=2}^{t} \log \Big(\frac{q(\mathbf{x}_{i-1}|\mathbf{x}_i,\mathbf{x}_0)}{p_\theta(\mathbf{x}_{i-1}|\mathbf{x}_i)}\cdot\frac{q(\mathbf{x}_i|\mathbf{x}_0)}{q(\mathbf{x}_{i-1}|\mathbf{x}_0)}\Big) + \log \frac{q(\mathbf{x}_1|\mathbf{x}_0)}{p_\theta(\mathbf{x}_0|\mathbf{x}_1)}\Big] \\
&= \mathbb{E}_q\Big[-\log p_\theta(\mathbf{x}_t) + \sum_{i=2}^{t} \log \frac{q(\mathbf{x}_{i-1}|\mathbf{x}_i,\mathbf{x}_0)}{p_\theta(\mathbf{x}_{i-1}|\mathbf{x}_i)} + \sum_{i=2}^{t} \log \frac{q(\mathbf{x}_i|\mathbf{x}_0)}{q(\mathbf{x}_{i-1}|\mathbf{x}_0)} + \log \frac{q(\mathbf{x}_1|\mathbf{x}_0)}{p_\theta(\mathbf{x}_0|\mathbf{x}_1)}\Big] \\
&= \mathbb{E}_q\Big[-\log p_\theta(\mathbf{x}_t) + \sum_{i=2}^{t} \log \frac{q(\mathbf{x}_{i-1}|\mathbf{x}_i,\mathbf{x}_0)}{p_\theta(\mathbf{x}_{i-1}|\mathbf{x}_i)} + \log \frac{q(\mathbf{x}_t|\mathbf{x}_0)}{q(\mathbf{x}_1|\mathbf{x}_0)} + \log \frac{q(\mathbf{x}_1|\mathbf{x}_0)}{p_\theta(\mathbf{x}_0|\mathbf{x}_1)}\Big] \\
&= \mathbb{E}_q\Big[\log \frac{q(\mathbf{x}_t|\mathbf{x}_0)}{p_\theta(\mathbf{x}_t)} + \sum_{i=2}^{t} \log \frac{q(\mathbf{x}_{i-1}|\mathbf{x}_i,\mathbf{x}_0)}{p_\theta(\mathbf{x}_{i-1}|\mathbf{x}_i)} - \log p_\theta(\mathbf{x}_0|\mathbf{x}_1)\Big] \\
&= \mathbb{E}_q\big[\underbrace{D_{\mathrm{KL}}(q(\mathbf{x}_t|\mathbf{x}_0) \parallel p_\theta(\mathbf{x}_t))}_{L_t} + \sum_{i=2}^{t} \underbrace{D_{\mathrm{KL}}(q(\mathbf{x}_{i-1}|\mathbf{x}_i,\mathbf{x}_0) \parallel p_\theta(\mathbf{x}_{i-1}|\mathbf{x}_i))}_{L_{i-1}} \underbrace{-\log p_\theta(\mathbf{x}_0|\mathbf{x}_1)}_{L_0}\big]
\end{aligned}
\tag{11}
$$

For diffusion model, the $L_i$ term is parameterized to minimize the difference of these two distributions $q(\mathbf{x}_{t-1}|\mathbf{x}_t,\mathbf{x}_0)$ and $p_\theta(\mathbf{x}_{t-1}|\mathbf{x}_t)$. Since both distributions are gaussian distributions, the problem can be translated to minimize the difference of their mean, which can be further derived into the following formulation:

$$
\begin{aligned}
L_t &= D_{\mathrm{KL}}(q(\mathbf{x}_t|\mathbf{x}_0) \parallel p_\theta(\mathbf{x}_t)) = C_t \\
L_i &= \mathbb{E}_{\mathbf{x}_0,\epsilon}\Big[\frac{(1-\alpha_i)^2}{2\alpha_i(1-\bar{\alpha}_i)\|\mathbf{\Sigma}_i\|_2^2}\|\boldsymbol{\epsilon}_i - \boldsymbol{\epsilon}_\theta(\sqrt{\bar{\alpha}_i}\mathbf{x}_0 + \sqrt{1-\bar{\alpha}_i}\boldsymbol{\epsilon}_i, i)\|^2\Big], i \in [1, t-1] \\
L_0 &= -\log p_\theta(\mathbf{x}_0|\mathbf{x}_1)
\end{aligned}
\tag{12}
$$

Consequently, $\bar{L}_t$ can be finally formulated as:

$$
\bar{L}_t = C_t + \sum_{i=1}^{t-1}\Big[\frac{(1-\alpha_i)^2}{2\alpha_i(1-\bar{\alpha}_i)\|\mathbf{\Sigma}_i\|_2^2}E_i\Big] + L_0,
\tag{13}
$$

where $E_i = \|\boldsymbol{\epsilon}_i - \boldsymbol{\epsilon}_\theta(\sqrt{\bar{\alpha}_i}\mathbf{x}_0 + \sqrt{1-\bar{\alpha}_i}\boldsymbol{\epsilon}_i, i)\|^2$.

$\square$

## C   PROOF OF **THEOREM 1.** IN SEC. 3.4

*Proof.* As shown in Eq. (12), when we change the initial $\mathbf{x}_0$ from high-resolution image $\mathbf{x}_0$ to low-resolution image $\hat{\mathbf{x}}_0$, the main different come from the $L_t$ term. Since $p_\theta(\mathbf{x}_t)$ does not change, we will focus on the change of $q(\mathbf{x}_t|\mathbf{x}_0)$. Note that $q(\mathbf{x}_t|\mathbf{x}_0)$ is the forward distribution which we have:

$$q(\mathbf{x}_t|\mathbf{x}_0) = \mathcal{N}(\mathbf{x}_t; \sqrt{\bar{\alpha}_t}\mathbf{x}_0, (1 - \bar{\alpha}_t)\mathbf{I}) \tag{14}$$

To analyze the changes, we can analyze the DK-divergence of these two distributions $q(\mathbf{x}_t|\mathbf{x}_0)$ and $q(\mathbf{x}_t|\hat{\mathbf{x}}_0)$.

$$
\begin{aligned}
D_{\mathrm{KL}}&(q(\mathbf{x}_t|\mathbf{x}_0) \parallel q(\mathbf{x}_t|\hat{\mathbf{x}}_0)) \\
&= \int q(\mathbf{x}_t|\mathbf{x}_0) \ln(\frac{q(\mathbf{x}_t|\mathbf{x}_0)}{q(\mathbf{x}_t|\hat{\mathbf{x}}_0)})d\mathbf{x} \\
&= \int q(\mathbf{x}_t|\mathbf{x}_0) \ln(\frac{e^{-\frac{(\mathbf{x} - \sqrt{\bar{\alpha}_t}\mathbf{x}_0)^2}{(1 - \bar{\alpha}_t)\mathbf{I}}}}{e^{-\frac{(\mathbf{x} - \sqrt{\bar{\alpha}_t}\hat{\mathbf{x}}_0)^2}{(1 - \bar{\alpha}_t)\mathbf{I}}}})d\mathbf{x} \\
&= \int q(\mathbf{x}_t|\mathbf{x}_0)(-\frac{(\mathbf{x} - \sqrt{\bar{\alpha}_t}\mathbf{x}_0)^2}{(1 - \bar{\alpha}_t)\mathbf{I}} + \frac{(\mathbf{x} - \sqrt{\bar{\alpha}_t}\hat{\mathbf{x}}_0)^2}{(1 - \bar{\alpha}_t)\mathbf{I}})d\mathbf{x} \\
&= \frac{1}{(1 - \bar{\alpha}_t)\mathbf{I}} \int q(\mathbf{x}_t|\mathbf{x}_0)(\bar{\alpha}_t(\mathbf{x}_0)^2 + \bar{\alpha}_t(\hat{\mathbf{x}}_0)^2 + 2\sqrt{\bar{\alpha}_t}(\mathbf{x}_0 - \hat{\mathbf{x}}_0)\mathbf{x})d\mathbf{x} \\
&= \frac{1}{(1 - \bar{\alpha}_t)\mathbf{I}} \int q(\mathbf{x}_t|\mathbf{x}_0)(\bar{\alpha}_t(\mathbf{x}_0)^2 + \bar{\alpha}_t(\hat{\mathbf{x}}_0)^2)d\mathbf{x} \\
&\quad + \frac{1}{(1 - \bar{\alpha}_t)\mathbf{I}} \int q(\mathbf{x}_t|\mathbf{x}_0)2\sqrt{\bar{\alpha}_t}(\mathbf{x}_0 - \hat{\mathbf{x}}_0)\mathbf{x})d\mathbf{x} \\
&= \frac{\bar{\alpha}_t(\mathbf{x}_0)^2 + \bar{\alpha}_t(\hat{\mathbf{x}}_0)^2}{(1 - \bar{\alpha}_t)\mathbf{I}} \int q(\mathbf{x}_t|\mathbf{x}_0)d\mathbf{x} + \frac{2\sqrt{\bar{\alpha}_t}(\mathbf{x}_0 - \hat{\mathbf{x}}_0)}{(1 - \bar{\alpha}_t)\mathbf{I}} \int q(\mathbf{x}_t|\mathbf{x}_0)\mathbf{x}d\mathbf{x} \\
&= \frac{\bar{\alpha}_t(\mathbf{x}_0)^2 + \bar{\alpha}_t(\hat{\mathbf{x}}_0)^2}{(1 - \bar{\alpha}_t)\mathbf{I}} + \frac{2\sqrt{\bar{\alpha}_t}(\mathbf{x}_0 - \hat{\mathbf{x}}_0)}{(1 - \bar{\alpha}_t)\mathbf{I}} \int q(\mathbf{x}_t|\mathbf{x}_0)\mathbf{x}d\mathbf{x}
\end{aligned}
\tag{15}
$$

As $q(\mathbf{x}_t|\mathbf{x}_0)$ is a gaussian distribution, $\int q(\mathbf{x}_t|\mathbf{x}_0)\mathbf{x}d\mathbf{x}$ is exactly the mean of this distribution which is $\sqrt{\bar{\alpha}_t}\mathbf{x}_0$, so we have

$$D_{\mathrm{KL}}(q(\mathbf{x}_t|\mathbf{x}_0) \parallel q(\mathbf{x}_t|\hat{\mathbf{x}}_0)) = \frac{\bar{\alpha}_t(\mathbf{x}_0)^2 + \bar{\alpha}_t(\hat{\mathbf{x}}_0)^2}{(1 - \bar{\alpha}_t)\mathbf{I}} + \frac{2\sqrt{\bar{\alpha}_t}(\mathbf{x}_0 - \hat{\mathbf{x}}_0)}{(1 - \bar{\alpha}_t)\mathbf{I}}\sqrt{\bar{\alpha}_t}\mathbf{x}_0 \tag{16}$$

When we increase the noise injection level $s$, $\bar{\alpha}_t$ decreases, then we can find $D_{\mathrm{KL}}(q(\mathbf{x}_t|\mathbf{x}_0) \parallel q(\mathbf{x}_t|\hat{\mathbf{x}}_0))$ decreases which means $q(\mathbf{x}_t|\hat{\mathbf{x}}_0)$ becomes more similar with $q(\mathbf{x}_t|\mathbf{x}_0)$. Especially, when $t \to \infty$, $\bar{\alpha}_t \to 0$, then $D_{\mathrm{KL}}(q(\mathbf{x}_t|\mathbf{x}_0 \parallel q(\mathbf{x}_t|\hat{\mathbf{x}}_0)) = 0$, both $q(\mathbf{x}_t|\mathbf{x}_0)$ and $q(\mathbf{x}_t|\hat{\mathbf{x}}_0)$ follows the same gaussian distribution. Besides, we can also find the gap between these two distributions is linear to the error between the error $(\mathbf{x}_0 - \hat{\mathbf{x}}_0)$ and decreases along with the noise injection level $t$. Then the $L_t$ term with $\hat{\mathbf{x}}_0$ as input can be:

$$
\begin{aligned}
L_t &= D_{\mathrm{KL}}(q(\mathbf{x}_t|\hat{\mathbf{x}}_0) \parallel p_\theta(\mathbf{x}_t)) \\
&\triangleq D_{\mathrm{KL}}(q(\mathbf{x}_t|\mathbf{x}_0) \parallel p_\theta(\mathbf{x}_t)) + D_{\mathrm{KL}}(q(\mathbf{x}_t|\hat{\mathbf{x}}_0) \parallel q(\mathbf{x}_t|\mathbf{x}_0)) \\
&= L_t + A_t + K_t(\mathbf{x}_0 - \hat{\mathbf{x}}_0)
\end{aligned}
\tag{17}
$$

Where $L_t$ is the forward error when we use $\mathbf{x}$ as input, $A_t$ is the first term in Eq. (16), $K_t(\mathbf{x}_0 - \hat{\mathbf{x}}_0)$ is the second term. The second line is because we have proved that the KL-divergence between $q(\mathbf{x}_t|\mathbf{x}_0)$ and $q(\mathbf{x}_t|\hat{\mathbf{x}}_0)$ becomes smaller as $t$ increase, so these two distributions become more and more similar as $t$ increases. So we can use $q(\mathbf{x}_t|\hat{\mathbf{x}}_0))$ to approximate $q(\mathbf{x}_t|\mathbf{x}_0)$ to some extent. Then we have:

$$D_{\text{KL}}(q(\mathbf{x}_t|\mathbf{x}_0) \parallel p_\theta(\mathbf{x}_t)) + D_{\text{KL}}(q(\mathbf{x}_t|\hat{\mathbf{x}}_0) \parallel q(\mathbf{x}_t|\mathbf{x}_0))$$

$$= \int q(\mathbf{x}_t|\mathbf{x}_0) \log \frac{q(\mathbf{x}_t|\mathbf{x}_0)}{p_\theta(\mathbf{x}_t)} + \int q(\mathbf{x}_t|\hat{\mathbf{x}}_0) \log \frac{q(\mathbf{x}_t|\hat{\mathbf{x}}_0)}{q(\mathbf{x}_t|\mathbf{x}_0)}$$

$$\triangleq \int q(\mathbf{x}_t|\hat{\mathbf{x}}_0) \log \frac{q(\mathbf{x}_t|\mathbf{x}_0)}{p_\theta(\mathbf{x}_t)} + \int q(\mathbf{x}_t|\hat{\mathbf{x}}_0) \log \frac{q(\mathbf{x}_t|\hat{\mathbf{x}}_0)}{q(\mathbf{x}_t|\mathbf{x}_0)} \quad (18)$$

$$= \int q(\mathbf{x}_t|\hat{\mathbf{x}}_0) \big( \log \frac{q(\mathbf{x}_t|\mathbf{x}_0)}{p_\theta(\mathbf{x}_t)} + \log \frac{q(\mathbf{x}_t|\hat{\mathbf{x}}_0)}{q(\mathbf{x}_t|\mathbf{x}_0)} \big)$$

$$= \int q(\mathbf{x}_t|\hat{\mathbf{x}}_0) \log \frac{q(\mathbf{x}_t|\mathbf{x}_0)}{p_\theta(\mathbf{x}_t)}$$

Consequently, the final recovery error $\mathcal{L}_t$ can be revised as:

$$\mathcal{L}_t = L_t + \sum_{i=0}^{t-1} L_i$$

$$= D_{\text{KL}}(q(\mathbf{x}_t|\hat{\mathbf{x}}_0) \parallel p_\theta(\mathbf{x}_t)) + \sum_{i=0}^{t-1} L_i$$

$$\triangleq D_{\text{KL}}(q(\mathbf{x}_t|\mathbf{x}_0) \parallel p_\theta(\mathbf{x}_t)) + D_{\text{KL}}(q(\mathbf{x}_t|\hat{\mathbf{x}}_0) \parallel q(\mathbf{x}_t|\mathbf{x}_0)) + \sum_{i=0}^{t-1} L_i \quad (19)$$

$$= \sum_{i=0}^{t} L_i + \big[ K_t(\mathbf{x}_0 - \hat{\mathbf{x}}_0) + A_t \big]$$

$$\triangleq \bar{L}_t + \omega \big[ K_t \parallel \mathbf{x} - \hat{\mathbf{x}}_0 \parallel^2 + A_t \big]$$

$$\triangleq \underbrace{\mathcal{L}_t^S}_{\text{Signature Loss}} + \underbrace{\omega \mathcal{L}_t^F}_{L_1 \text{ Fidelity Loss}}$$

where $\omega$ is a hyperparameter to guarantee these two loss terms are of the same magnitude, we choose $\omega = 0.004$.

$\square$

## D   DETAILS OF EXPERIMENTAL SETUP IN SEC. 4.1

**Baselines and Datasets.** We use Bicubic (Aràndiga, 2016; Hwang & Lee, 2004), Nearest (Lin et al., 2008), EDSR (Lim et al., 2017), ESRGAN (Wang et al., 2018), LIIF (Chen et al., 2021), and SR3 (Saharia et al., 2021) as the baseline solutions for performance comparison.

To guarantee evaluation fairness, the models of baseline solutions, as well as the pre-trained DGM used by Diff-SR, are established on a unified experimental setup. We use four pertinent SR datasets, including DIV2K (Agustsson & Timofte, 2017), Set5 (Bevilacqua et al., 2012), Set14 (Zeyde et al., 2010), and Urban100 (Huang et al., 2015). Following the pre-training guidances mentioned in the corresponding work of the baseline solutions, we use the high-resolution (HR) images as the supervised ground truth to pre-train the baseline models, so that they can recover the low-resolution (LR) images to the HR versions. Considering the property of the diffusion process, the DGM only requires the HR images to optimize its restoration capacity, without access need for the LR images.

**Setting of Pre-trained DGM.** The DGM used by Diff-SR employs the UNet backbone (Ronneberger, 2017) and follows the sampling strategy of *Denoising Diffusion Implicit Models* (DDIMs) (Song et al., 2021a). More precisely, the DGM uses three downsampling blocks, two middle blocks and three upsampling blocks in UNet. The scaling factors are set as 2, 4, 8 for these three kinds of blocks, respectively. The number of base feature channels is 64. To capture the time sequence information, the diffusion step index $t$ is specified by adding the sinusoidal position embedding into each residual block. By setting the maximum step number as $T = 1000$, the DGM controls the noise variance

$\beta_t$ ($t \in [1, T]$) through a linear quadratic scheduler, which gradually ranges from $\beta_1 = 10^{-4}$ to $\beta_T = 0.02$. Also, the DGM is optimized by the *Mean of Squared Error* (MSE) loss with Adam optimizer (Kingma & Ba, 2015) and 16 batch size. The total number of epochs is 10K and the initial learning rate is $1 \times 10^{-5}$.

**Test setting.** Our Diff-SR and the other baseline solutions take the LR images to generate the recovered HR images. The performance is compared by checking the scores of FID, SSIM and PSNR, between the recovered HR images and the ground-truth HR ones.

## E  DEEP INSPECTION FROM FREQUENCY DOMAIN

Another interesting result is when we analyze this problem in the frequency domain (through Fourier transform (Bracewell & Bracewell, 1986; Brigham & Morrow, 1967)), it gives us another perspective to understand why this noise injection operation can restore a low-resolution image to a high-resolution one. As shown in Figure. 8, we calculate the model output $\tilde{\mathbf{x}}$'s frequency map and compare it with the frequency map of the original image $\mathbf{x}$, we use two boxes to segment the low-frequency information and high-frequency information as shown in the supplementary material of §E. Besides, we also illustrate the frequency map of the intermediate results that we get after injecting noise, which is shown in the lower side of Figure. 8.

Different images in the spatial and frequency domains are shown in the first box at the initial state. Compared with the original image $\mathbf{x}_0$, the low-resolution image lost most of the high-frequency information (*i.e.,* four corners) and the low-frequency information is preserved. Then when we inject some noise into these images, their frequency maps also change. But as shown in the second box, if the noise injection level is insufficient, there are still some significant differences. As a result, when we compare the enhanced result $\tilde{\mathbf{x}}$, $\tilde{\mathbf{x}} \sim q(\mathbf{x}|\hat{\mathbf{x}}_t)$ with ground truth $\mathbf{x}_0$, we can find the low-frequency error is small but the high-frequency error is large. From the spatial domain, this means the content of the output image is consistent with the ground truth, but the image is blurred. Then if we inject enough noise (*i.e.,* in the third box), the frequency domain maps are very similar. We can get a clear output image when we denoise this image with DGM. Both high-frequency and low-frequency errors are less than a threshold, so the output image is clear and content-right. If we inject too much noise as shown in the final box, the frequency maps are similar, but the low-frequency information is destroyed. Then when we start the reverse process at this data point, the output image suffers from high low-frequency error even though the low-frequency error is small. From the spatial domain, this means we have a clear output image but the semantic content of this image is different from the original image.

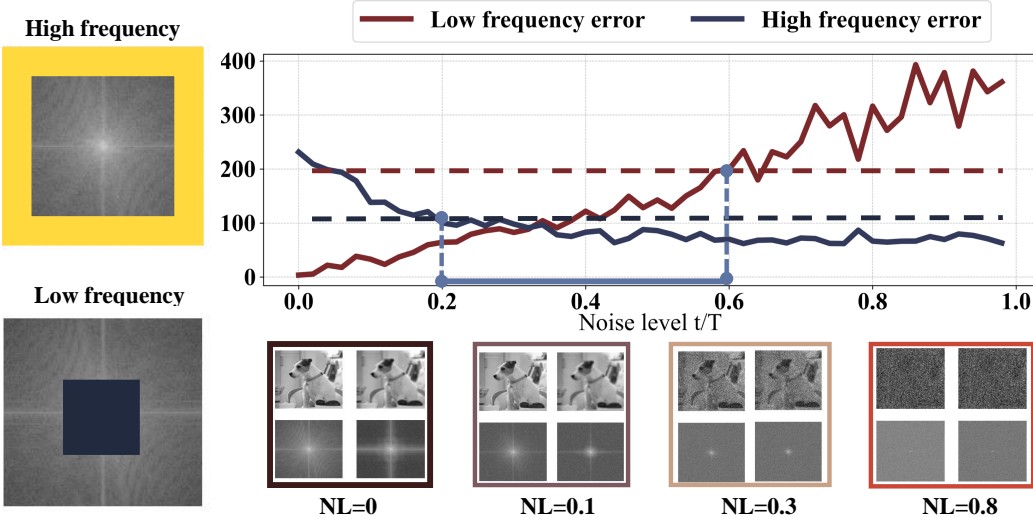

Figure 8: Illustration on frequency domain. For each box in the figure, the upper left is the result of the **original image** inject $t$ steps noise, the lower left is the frequency map of this image, the upper right is the result of **low-resolution image** inject $t$ steps noise, lower right is the frequency map.

# F  QUALITATIVE RESULTS IN SEC. 4.2

In this section, we provide the qualitative results on the datasets of DIV2K (Agustsson & Timofte, 2017), Set5 (Bevilacqua et al., 2012), Set14 (Zeyde et al., 2010), and Urban100 (Huang et al., 2015). From Figure. 9 to Figure. 20, we can observe that Diff-SR consistently achieves higher recovery quality over the baseline solutions.

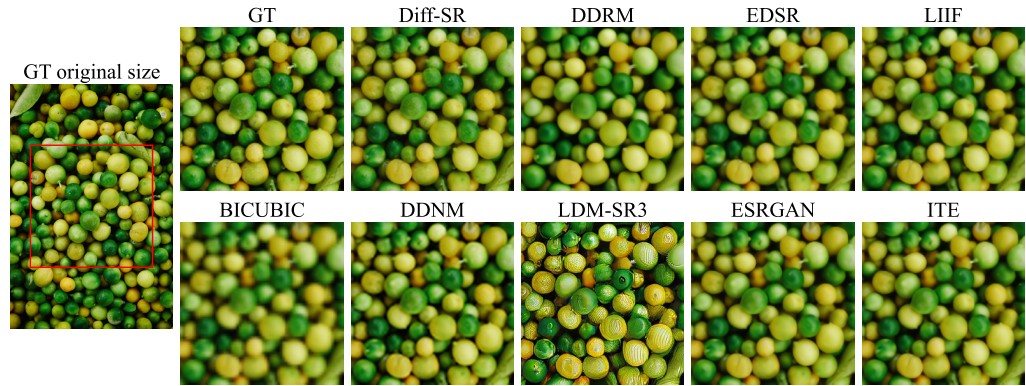

Figure 9: Qualitative results of different $4\times$ SR solutions, evaluated on DIV2K dataset.

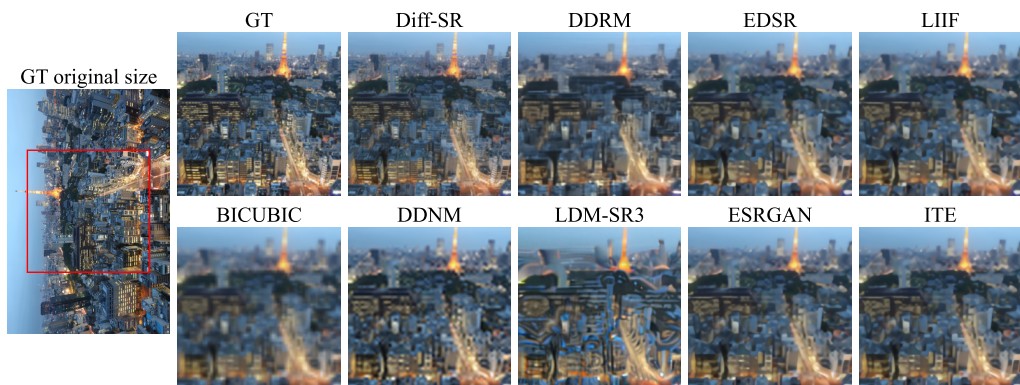

Figure 10: Qualitative results of different $4\times$ SR solutions, evaluated on DIV2K dataset.

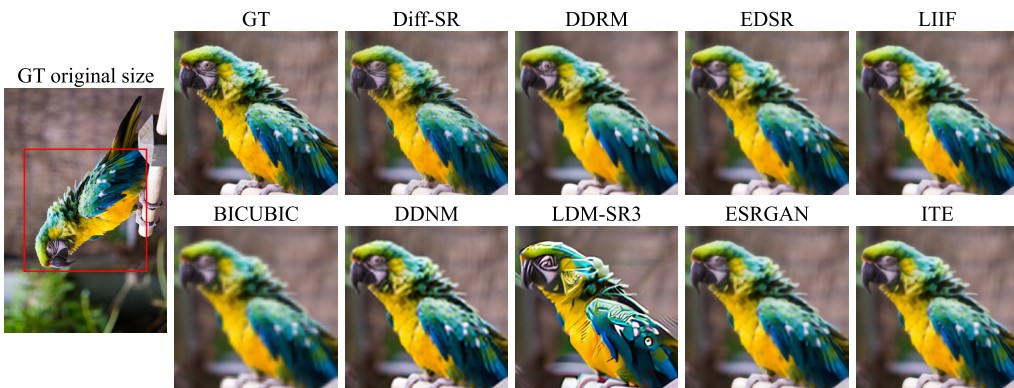

Figure 11: Qualitative results of different $4\times$ SR solutions, evaluated on DIV2K dataset.

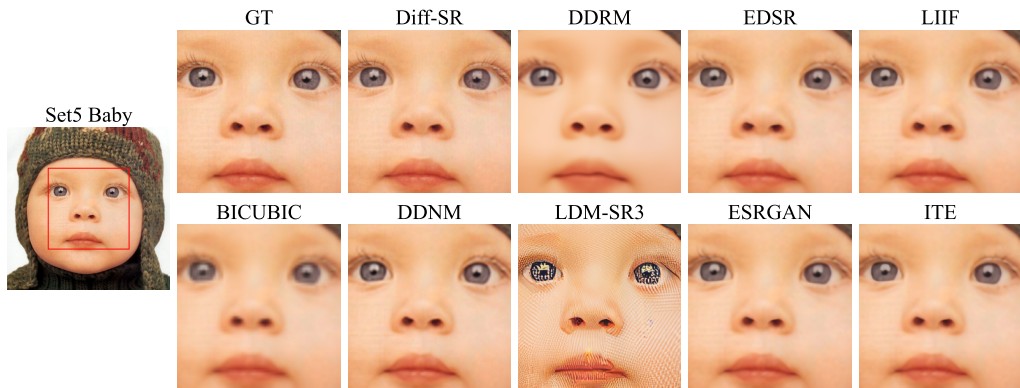

Figure 12: Qualitative results of different $4\times$ SR solutions, evaluated on Set5 dataset.

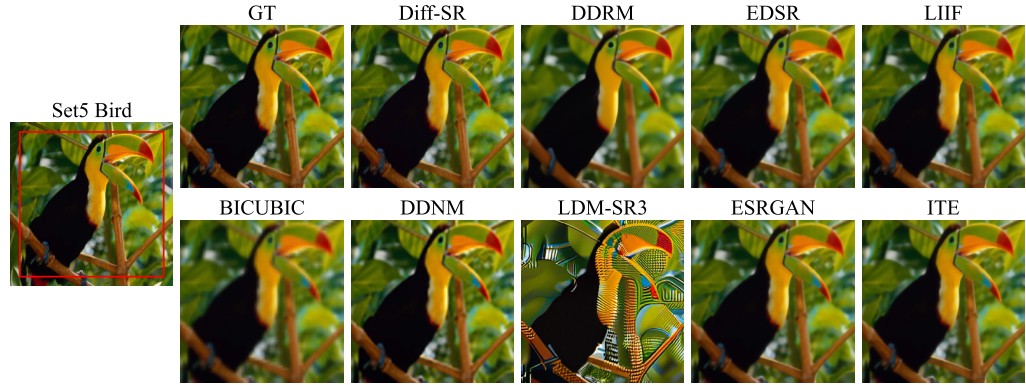

Figure 13: Qualitative results of different $4\times$ SR solutions, evaluated on Set5 dataset.

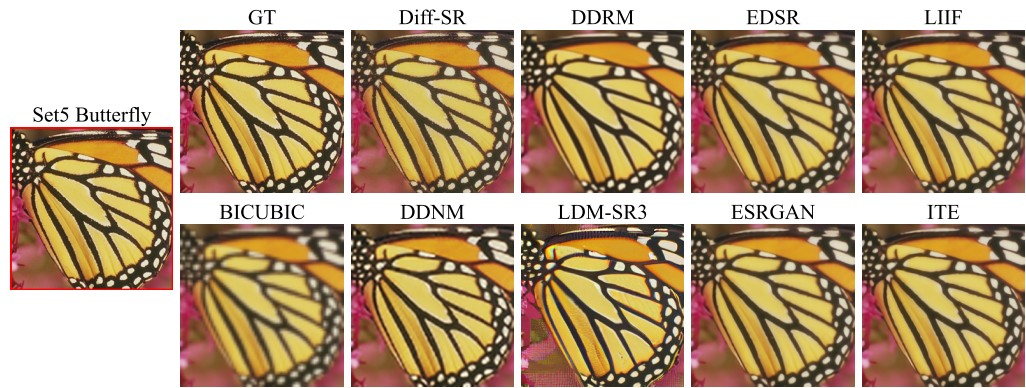

Figure 14: Qualitative results of different $4\times$ SR solutions, evaluated on Set5 dataset.

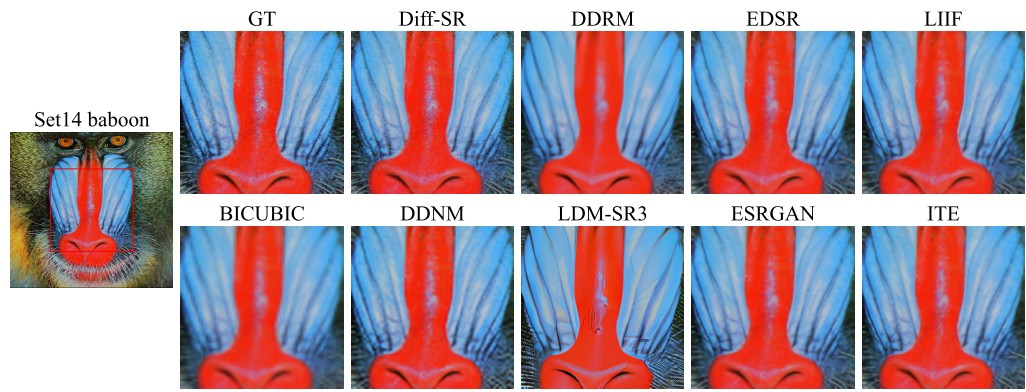

Figure 15: Qualitative results of different $4\times$ SR solutions, evaluated on Set14 dataset.

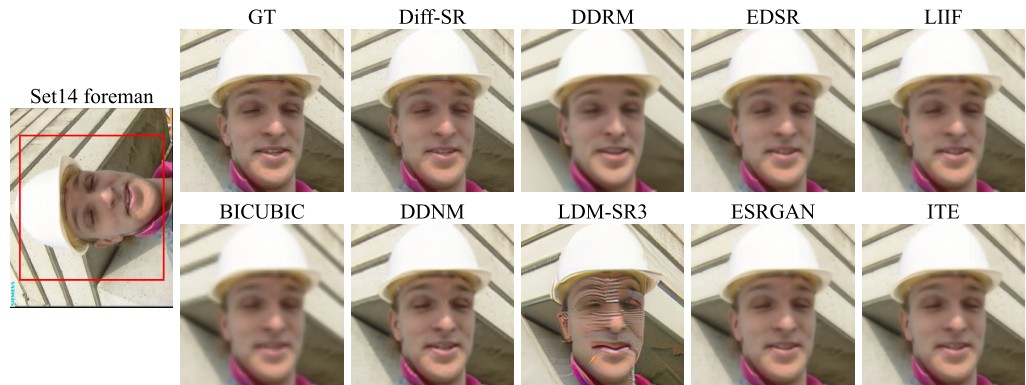

Figure 16: Qualitative results of different $4\times$ SR solutions, evaluated on Set14 dataset.

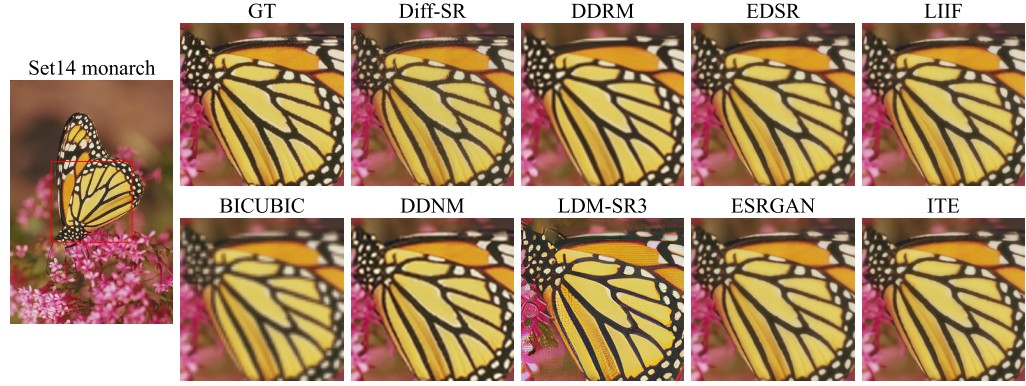

Figure 17: Qualitative results of different $4\times$ SR solutions, evaluated on Set14 dataset.

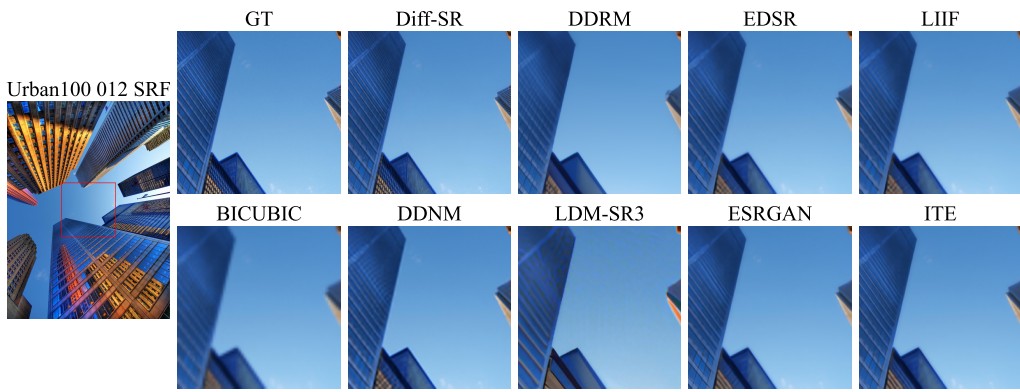

Figure 18: Qualitative results of different $4\times$ SR solutions, evaluated on Urban100 dataset.

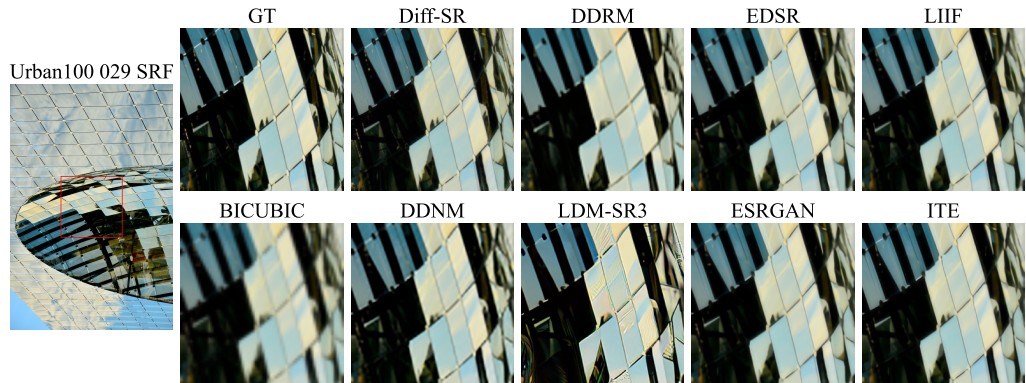

Figure 19: Qualitative results of different $4\times$ SR solutions, evaluated on Urban100 dataset.

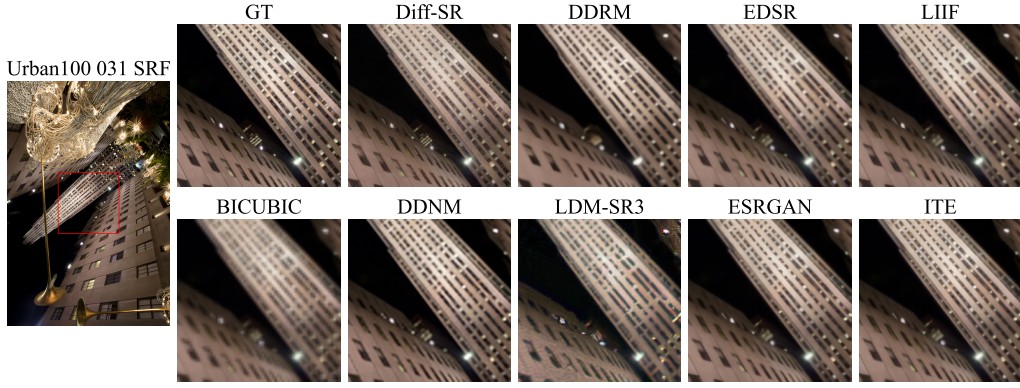

Figure 20: Qualitative results of different $4\times$ SR solutions, evaluated on Urban100 dataset.

