# SUPPLEMENTARY VISUAL COMPARISON

## A    EVALUATION ON NOVEL BASELINES

It is important to note that the experimental comparison may not be entirely fair, as StableSR can only upscale LR images to 512, while our experimental settings require a target pixel size of 256x256. Therefore, we had to downsample the StableSR image to match our target size. We provide this result as a reference, understanding that the comparison may not be entirely equitable.

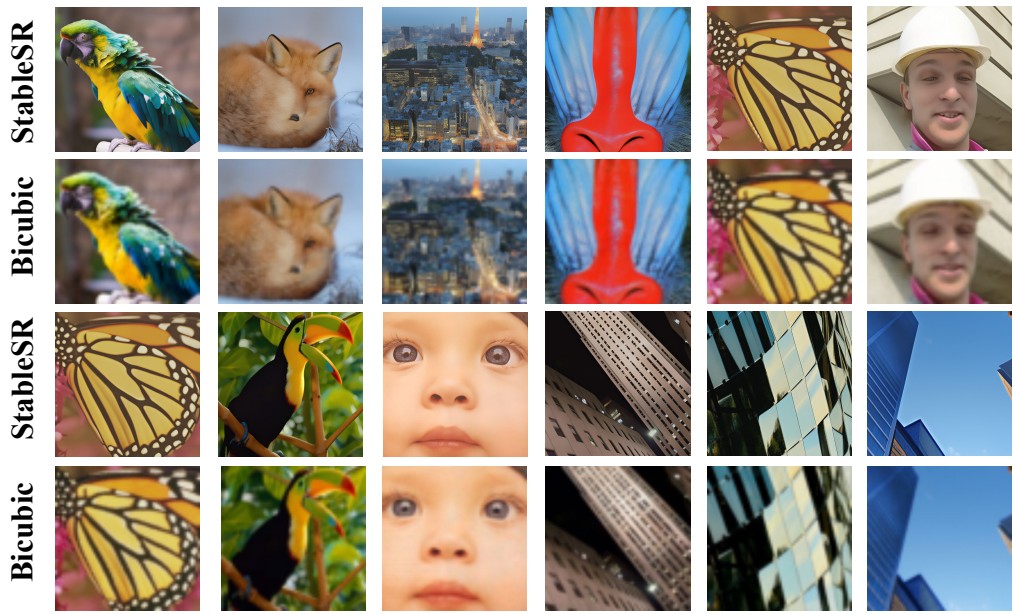

Figure 1: Qualitative results of StableSR.

## B    VISUAL COMPARISON ON DEGRADATION

We followed the degradation pipeline designed by ESRGAN and tested it using the corresponding degradation code on DIV2K. We applied Blur, JPEG compression, and noise methods to degrade the original image. We then injected noise and denoised the image to evaluate the recovery effect of the model. Additionally, we used the two-order pipe from ESRGAN-pipe to degrade the image multiple times and finally reduced the image by 2 times. Interestingly, when using only a single type of degradation, the images were still repaired effectively. Among the mentioned degradations, the model performed best in recovering blur, but struggled the most with compression at 4x scale. Furthermore, we observed that the degradation results of image processing with a single degradation method were not visually obvious, especially in the case of JPEG compression. However, when applying ESRGAN's pipeline to degrade images multiple times, the model's performance significantly declined. Therefore, image recovery with multiple degradations remains a challenging task.

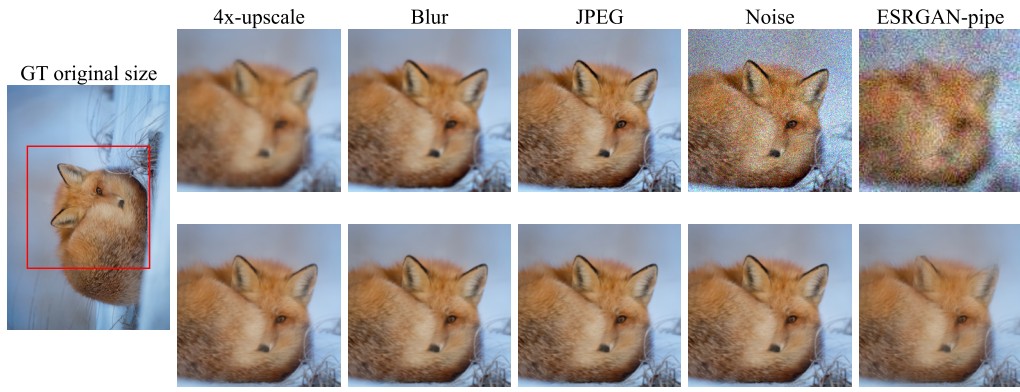

Figure 2: Qualitative results of different degradation, evaluated on DIV2K dataset.

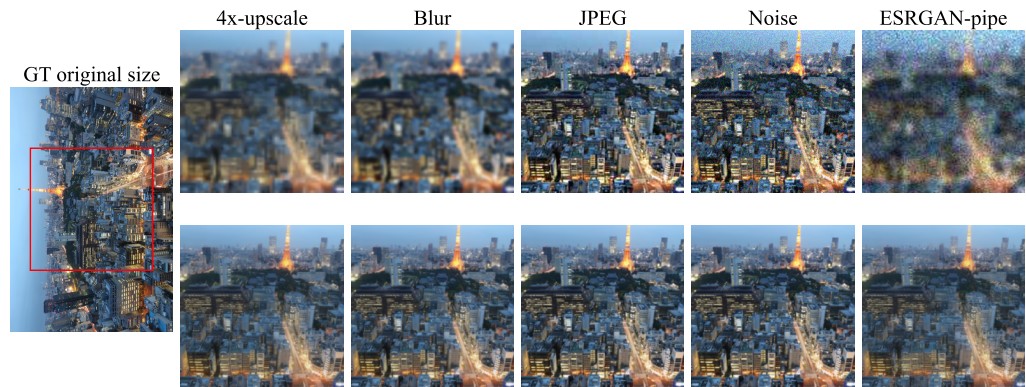

Figure 3: Qualitative results of different degradation, evaluated on DIV2K dataset.

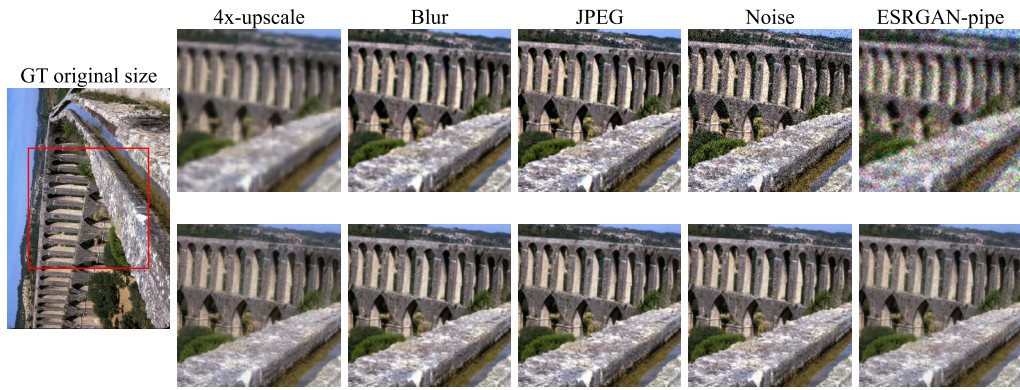

Figure 4: Qualitative results of different degradation, evaluated on DIV2K dataset.

# C  VISUAL COMPARISON ON NO-INTEGER SCALE SR

As illustrated in Figure 2(b), images of different scales almost follow the same path to return HR images, so there is no big visual change between different scale.

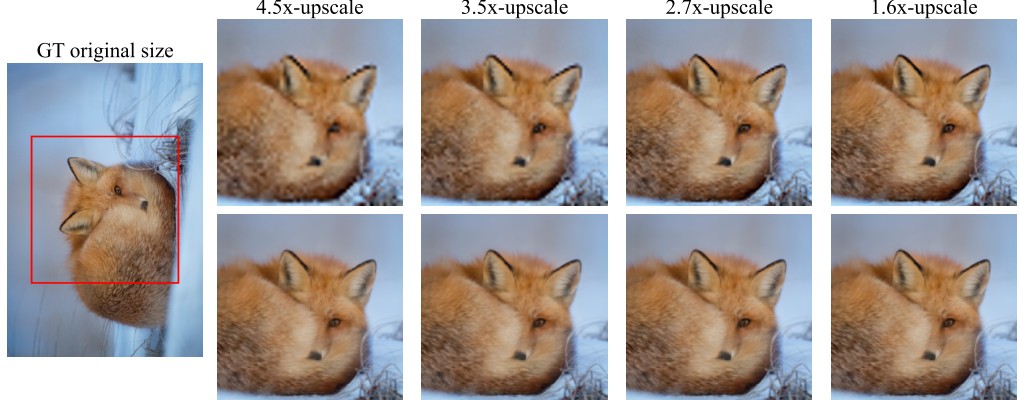

Figure 5: Qualitative results of different scale, evaluated on DIV2K dataset.

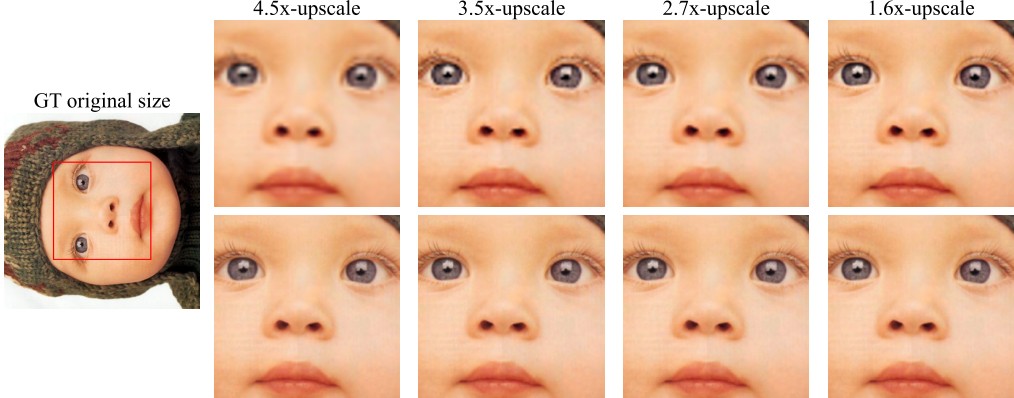

Figure 6: Qualitative results of different scale, evaluated on Set5 dataset.

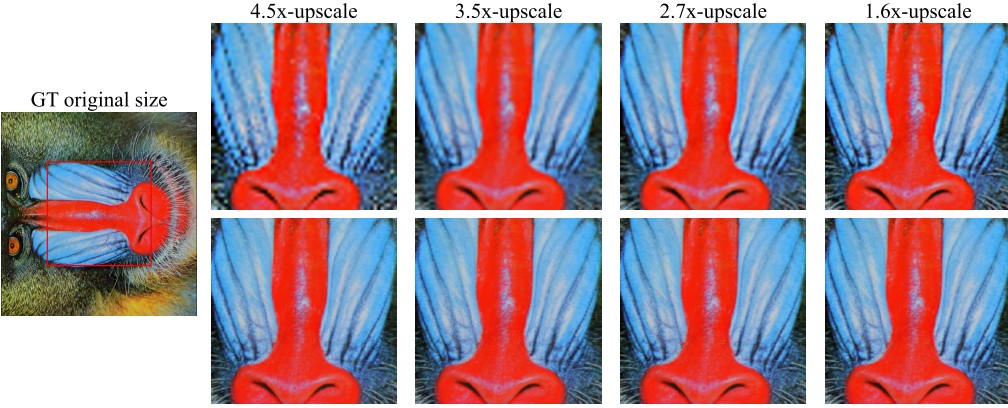

Figure 7: Qualitative results of different scale, evaluated on Set14 dataset.

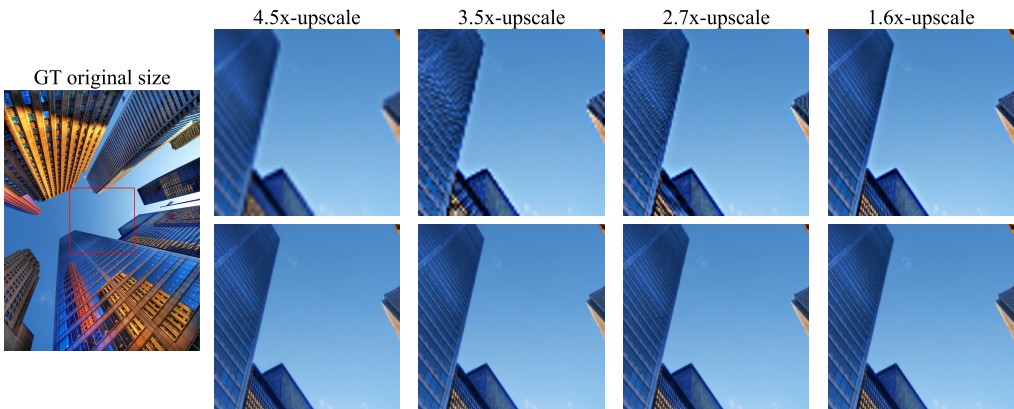

Figure 8: Qualitative results of different scale, evaluated on Urban100 dataset.