# OpenReview forum: "Dissecting Arbitrary-scale Super-resolution Capability from Pre-trained Diffusion Generative Models"
_ICLR.cc/2024/Conference — Submitted to ICLR 2024_

### Official Review · Reviewer_aha2 · 2023-10-24

**Soundness:** 3 good
**Presentation:** 3 good
**Contribution:** 3 good
**Rating:** 6
**Confidence:** 4

**Summary:**

In this paper, the authors introduce a new paradigm to achieve scale-arbitrary SR by exploiting pre-trained diffusion generative models without any additional training overhead. Specifically, a specific amount of noise is first injected into the low-resolution images and then the resultant images are fed to diffusion generative models to generate the SR results. The authors present theoretical analyses and propose a perceptual recoverable field to achieve the trade-off between the fidelity and the realness. Extensive experiments show that the proposed method produces state-of-the-art performance on several benchmark datasets.

**Strengths:**

+ This paper is well-motivated and well-written.
+ Endowing existing diffusion generative models with the capablity of scale-arbitrary SR without additional overhead is interesting.

**Weaknesses:**

- As this paper aims to dissect arbitrary-scale SR capability, experiments should be conducted on arbitrary-scale SR tasks other than integer and symmetric scale factors (i.e., assymmetric SR).
- If I understand correctly, the proposed method is not only limited to arbitrary-scale SR tasks, but can be extended to arbitrary-degradation SR tasks. In my opinion, LR images with different scale factors are referred to as images with different degradation models such that different amounts of noise are required to recover the missing details. Consequently, I wonder more results on arbitrary-degradation SR.
- For fair comparison with previous perception-oriented methods, metrics like LPIPS should be included for evaluation.
- As diffusion generative models require several iterations to produce promising results and usually computationally expensive, inference time should also be reported such that the readers can be clearly aware of the cost of the proposed method.

**Questions:**

please see weaknesses

---

> ### Author Response · Authors · 2023-11-16
>
> Q1: If I understand correctly, you are suggesting that we should include the performance results for non-integer scaling. We have included the results of our model as well as several baseline models. However, it is worth noting that most of these baselines only support integer-scale super-resolution. Therefore, we have only included two additional baselines, namely LIIF and ITSRN.
>
> | Scale | DIV2K-FID | DIV2K-PSNR | DIV2K-SSIM | Set5-FID | Set5-PSNR | Set5-SSIM | Set14-FID | Set14-PSNR | Set14-SSIM | Urban100-FID | Urban100-PSNR | Urban100-SSIM |
> | ---- | ---- | ---- | ---- | ---- | ---- | ---- | ---- | ---- | ---- | ---- | ---- | ---- |
> | DiffSR-4.5x | 0.624 | 33.098 | 0.938 | 0.117 | 36.628 | 0.967 | 0.185 | 35.822 | 0.967 | 0.390 | 33.200 | 0.941 |
> | ITSRN-4.5x | 3.327 | 29.645 | 0.625 | 3.268 | 31.125 | 0.762 | 3.212 | 29.856 | 0.655 | 4.842 | 29.519 | 0.593 |
> | LIIF-4.5x | 8.165 | 30.308 | 0.697 | 3.456 | 31.777 | 0.814 | 8.667 | 30.685 | 0.713 | 9.422 | 29.916 | 0.639 |
> | DiffSR-3.5x | 0.385 | 33.437 | 0.952 | 0.089 | 36.785 | 0.972 | 0.112 | 36.806 | 0.974 | 0.186 | 34.103 | 0.965 |
> | ITSRN-3.5x | 4.600 | 29.839 | 0.708 | 3.342 | 31.487 | 0.821 | 2.891 | 30.205 | 0.727 | 6.189 | 29.708 | 0.671 |
> | LIIF-3.5x | 4.392 | 30.613 | 0.775 | 2.364 | 32.083 | 0.858 | 5.639 | 31.047 | 0.778 | 5.799 | 30.122 | 0.718 |
> | DiffSR-2.7x | 0.257 | 33.405 | 0.957 | 0.076 | 36.405 | 0.976 | 0.074 | 36.900 | 0.975 | 0.120 | 34.897 | 0.975 |
> | ITSRN-2.7x | 2.795 | 30.150 | 0.805 | 2.957 | 31.794 | 0.871 | 2.851 | 30.859 | 0.828 | 4.106 | 29.906 | 0.768 |
> | LIIF-2.7x | 2.517 | 31.013 | 0.857 | 1.464 | 32.393 | 0.902 | 3.545 | 31.502 | 0.847 | 3.450 | 30.414 | 0.807 |
> | DiffSR-1.6x | 0.135 | 32.900 | 0.958 | 0.066 | 35.770 | 0.977 | 0.052 | 36.383 | 0.976 | 0.084 | 34.706 | 0.978 |
> | ITSRN-1.6x | 1.278 | 30.942 | 0.926 | 0.787 | 32.294 | 0.933 | 1.132 | 31.637 | 0.927 | 2.188 | 30.396 | 0.897 |
> | LIIF-1.6x | 0.840 | 31.569 | 0.938 | 0.979 | 32.605 | 0.939 | 1.085 | 32.137 | 0.930 | 1.120 | 30.957 | 0.914 |
>
> Q2: To evaluate our model, we followed the degradation pipeline designed by ESRGAN and used the corresponding degradation code. We applied various methods such as blur, JPEG compression, and noise to degrade the original image. We then injected noise and performed denoising to assess the recovery effect of our model. Additionally, we utilized the two-order pipeline from ESRGAN-pipe to degrade the image multiple times and finally downsampled it by a factor of 2. Interestingly, when using a single type of degradation, the model showed impressive image restoration capabilities. Among the different degradation methods, the model performed best in terms of recovering blurred images, but struggled the most with compression at a 4x scale. Furthermore, we observed that the visual degradation caused by a single degradation method, especially JPEG compression, was not very apparent. However, in real-world scenarios where images undergo multiple degradations using ESRGAN's pipeline, the model's performance significantly declined. Image restoration with multiple degradations remains a challenging task.
>
> | Method | DIV2K-FID | DIV2K-PSNR | DIV2K-SSIM | Set5-FID | Set5-PSNR | Set5-SSIM | Set14-FID | Set14-PSNR | Set14-SSIM | Urban100-FID | Urban100-PSNR | Urban100-SSIM |
> | ---- | ---- | ---- | ---- | ---- | ---- | ---- | ---- | ---- | ---- | ---- | ---- | ---- |
> | 4x | 1.088 | 32.450 | 0.924 | 0.196 | 34.732 | 0.961 | 0.281 | 34.982 | 0.964 | 0.915 | 32.148 | 0.910 |
> | Blur | 0.331 | 33.583 | 0.953 | 0.169 | 36.786 | 0.969 | 0.099 | 37.162 | 0.975 | 0.213 | 34.297 | 0.958 |
> | JPEG Compression | 0.127 | 32.295 | 0.954 | 0.058 | 35.001 | 0.975 | 0.046 | 35.649 | 0.974 | 0.106 | 33.711 | 0.974 |
> | Noise | 0.104 | 32.220 | 0.947 | 0.057 | 34.914 | 0.973 | 0.053 | 35.311 | 0.969 | 0.080 | 33.543 | 0.968 |
> | ESRGAN | 1.528 | 30.817 | 0.821 | 0.333 | 31.996 | 0.924 | 0.489 | 32.908 | 0.909 | 1.292 | 30.732 | 0.801 |
>
> Q3: Following your suggestion, we have included the results of LPIPS and the inference time, which are presented below:
>
> | Model | Scale | Time | DIV2K-LPIPS | Set5-LPIPS | Set14-LPIPS | Urban100-LPIPS |
> | ---- | ---- | ---- | ---- | ---- | ---- | ---- |
> |Bicubic | $4\times$ | 8ms | 0.412 | 0.278 | 0.376 | 0.441 |
> |DDNM | $4\times$ | 2s | 0.269 | 0.142 | 0.247 | 0.259 |
> |DDRM | $4\times$ | 2s | 0.271 | 0.195 | 0.290 | 0.246 |
> |LDM-SR3 | $4\times$ | 3s | 0.368 | 0.432 | 0.379 | 0.269 |
> |EDSR | $4\times$ | 15ms | 0.216 | 0.124 | 0.229 | 0.192 |
> |ESRGAN | $4\times$ | 20ms | 0.204 | 0.115 | 0.225 | 0.167 |
> |ite | $4\times$ | 155ms | 0.196 | 0.110 | 0.218 | 0.155 |
> |ITSRN | $4\times$ | 180ms | 0.228 | 0.128 | 0.231 | 0.216 |
> |LIIF | $4\times$ | 19ms | 0.211 | 0.117 | 0.226 | 0.188 |
> |SwinIR | $4\times$ | 465ms | 0.194 | 0.110 | 0.219 | 0.151 |
> |DiffSR | $4\times$ | 600ms | 0.125 | 0.092 | 0.096 | 0.128 |

---

### Official Review · Reviewer_oyvV · 2023-10-26

**Soundness:** 3 good
**Presentation:** 3 good
**Contribution:** 3 good
**Rating:** 5
**Confidence:** 5

**Summary:**

This paper proposed Diff-SR for arbitrary-scale super-resolution (ASSR) using one pre-trained Diffusion model without any additional training efforts. To achieve arbitrary-scale super-resolution, this paper proposed to add a specific amount of noise to LR images before the Diffusion model backward diffusion process. Besides, a metric Perceptual Recoverable Field has been proposed for achieving the optimal trade-off between fidelity and realness. Overall, this paper is well-written and easy to understand.

**Strengths:**

1. This paper presented a new perspective that extends a pre-trained diffusion model into an arbitrary-scale super-resolution by adding a specific amount of noise to LR images before the diffusion model backward diffusion process. The idea is interesting.
2. This paper proposed a metric Perceptual Recoverable Field for achieving the optimal trade-off between fidelity and realness.

**Weaknesses:**

Please refer to the Questions.

**Questions:**

1. Since the proposed method is for arbitrary-scale super-resolution, it is important to provide the quantity results comparison of arbitrary-scale such as 1.6X, 2.7X, 3.5X, and 4.5X. Also, the visual comparison of different scales comparison of the proposed method on the same image is also important to evaluate this work.
2. It mentioned that the used pre-trained diffusion model is 256*256. How does this model achieve arbitrary-scale super-resolution? For example, the low-resolution image is sized 100*100, how can we achieve 3.5X super-resolution?
3. Please give a reference for the value of noise level in an interval of 0.1.
4. Can the proposed method extend in real-world arbitrary-scale super-resolution?
5. Adding different noise levels when dealing with different scales, the higher noise means lower fidelity. When facing larger scale super-resolution, that means very low fidelity. Is there a big visual change between X2 and X4 results?
6. How does the proposed method maintain the trade-off between fidelity and realness?
7. In Line 22 of Page 2, this paper mentions high-level signature of generated content. What does it mean?
8. In Line 3 of Page 5, should the low-resolution image be high-resolution image? Otherwise, it is hard to understand this sentence.

---

> ### Author Response · Authors · 2023-11-16
>
> Q1: Thank you for your advice. We have included the results of ASSR baselines (LIIF, ITSRN). However, other baselines like SwinIR, ESRGAN, DDNM, etc., only support super-resolution with integer scales. Although we can compare them by first using an integer-scale model to upscale the images and then rescaling them to 256x256 using a method like Bicubic, we have been advised that this type of comparison is unfair.
>
> | Scale | DIV2K-FID | DIV2K-PSNR | DIV2K-SSIM | Set5-FID | Set5-PSNR | Set5-SSIM | Set14-FID | Set14-PSNR | Set14-SSIM | Urban100-FID | Urban100-PSNR | Urban100-SSIM |
> | ---- | ---- | ---- | ---- | ---- | ---- | ---- | ---- | ---- | ---- | ---- | ---- | ---- |
> | DiffSR-4.5x | 0.624 | 33.098 | 0.938 | 0.117 | 36.628 | 0.967 | 0.185 | 35.822 | 0.967 | 0.390 | 33.200 | 0.941 |
> | ITSRN-4.5x | 3.327 | 29.645 | 0.625 | 3.268 | 31.125 | 0.762 | 3.212 | 29.856 | 0.655 | 4.842 | 29.519 | 0.593 |
> | LIIF-4.5x | 8.165 | 30.308 | 0.697 | 3.456 | 31.777 | 0.814 | 8.667 | 30.685 | 0.713 | 9.422 | 29.916 | 0.639 |
> | DiffSR-3.5x | 0.385 | 33.437 | 0.952 | 0.089 | 36.785 | 0.972 | 0.112 | 36.806 | 0.974 | 0.186 | 34.103 | 0.965 |
> | ITSRN-3.5x | 4.600 | 29.839 | 0.708 | 3.342 | 31.487 | 0.821 | 2.891 | 30.205 | 0.727 | 6.189 | 29.708 | 0.671 |
> | LIIF-3.5x | 4.392 | 30.613 | 0.775 | 2.364 | 32.083 | 0.858 | 5.639 | 31.047 | 0.778 | 5.799 | 30.122 | 0.718 |
> | DiffSR-2.7x | 0.257 | 33.405 | 0.957 | 0.076 | 36.405 | 0.976 | 0.074 | 36.900 | 0.975 | 0.120 | 34.897 | 0.975 |
> | ITSRN-2.7x | 2.795 | 30.150 | 0.805 | 2.957 | 31.794 | 0.871 | 2.851 | 30.859 | 0.828 | 4.106 | 29.906 | 0.768 |
> | LIIF-2.7x | 2.517 | 31.013 | 0.857 | 1.464 | 32.393 | 0.902 | 3.545 | 31.502 | 0.847 | 3.450 | 30.414 | 0.807 |
> | DiffSR-1.6x | 0.135 | 32.900 | 0.958 | 0.066 | 35.770 | 0.977 | 0.052 | 36.383 | 0.976 | 0.084 | 34.706 | 0.978 |
> | ITSRN-1.6x | 1.278 | 30.942 | 0.926 | 0.787 | 32.294 | 0.933 | 1.132 | 31.637 | 0.927 | 2.188 | 30.396 | 0.897 |
> | LIIF-1.6x | 0.840 | 31.569 | 0.938 | 0.979 | 32.605 | 0.939 | 1.085 | 32.137 | 0.930 | 1.120 | 30.957 | 0.914 |
>
> Q2: In our arbitrary-scale super-resolution setting, we have a target resolution of 256x256. Our goal is to upscale any resolution under this setting to the target resolution. So, for an input of 100x100, we are actually conducting a 2.24x super-resolution. We consider this as an ASSR capacity, and we believe this setting can potentially be used in image and video applications. Since users prefer images and videos in fixed resolutions like 1080p, 2K, 4K, it may be a better choice to upscale an image with a resolution of 854x480 to 1920x1080 rather than upscaling it to a different scale with a similar resolution to 1920x1080.
>
> Q3: When we mention using the interval, we mean that if the total diffusion sampling steps from noise $x_T$ to image $x_0$ is 50 (T=50), we inject 0.1*T = 5 steps of noise to obtain $\hat{x}_t$ (t=5) and then denoise $\hat{x}_t$ to get $x_0$.
>
> Q4: As shown in Figure 3, the results obtained at different scales vary. For example, an acceptable result can be achieved at a noise level of 0.2 for a 2.5x magnification, but the 4.5x image still appears blurry at the same magnification. Therefore, we need to inject more noise. When the noise level reaches 0.3 or 0.4, the effect improves, but as we mentioned earlier, there may be slight differences in details compared to the original image, which we consider tolerable. The significant difference occurs when the noise level in the PRF exceeds the threshold of about 0.5-0.6, where a noticeable difference occurs. To strike a balance, our current strategy is to empirically estimate the appropriate recovery interval for each scale. As shown in Figure 4 and Figure 6, there is a suitable recovery interval for each scale. Within this interval, the model's recovery effect on the image does not vary significantly, so we choose the leftmost side of the interval as the noise level we inject.
>
> Q5: The low-level fidelity measure and high-level signature are consistent with Figure 3. In this context, the term "high-level signature" refers to whether the image depicts the same person, while "low-level fidelity" refers to the clarity or blurriness of the image.
>
> Q6: Thank you for pointing that out. In our statement, we actually meant to express the idea of "restoring a low-resolution image to a high-resolution image." We will carefully review our draft to avoid similar typos in the future.

---

> ### Author Response · Authors · 2023-11-21
>
> Sorry for the Q4 and Q5:
>
> For Q1 and Q2, we have revised our manuscript and change the claim from ASSR to TRSR (Target-resolution super-resolution).
>
> For Q3, here is the table of noise level that we use empirically. As we have proved in the appendix, The noise level is related to the difference between HR and LR images. Currently, we do not have a mathmatical tool to model this difference under different scale. So we leave it as our future work.
>
> | Scale | Noise Level |
> |-------|-------------|
> | 1.6x  | 0.1         |
> | 2x    | 0.1         |
> | 2.7x  | 0.2         |
> | 3.5x  | 0.3         |
> | 4x    | 0.5         |
> | 4.5x  | 0.5         |
>
> For Q4, we forget to post the answer as reviewer [cxpS] also ask the same question. The result is shown in the comments Q5 in the reviewer [cxpS]. Current, it is still challenging to restore the image in real-world complicated degradation setting. However, what we want to focus on in this paper is the ability to recover HR from any-scale LR image, we will keep improve the model performance under complicated real-world setting.
>
> For Q5, we have add the visual comparison in different scale in the supplementary file. Since images of different scales almost follow the same path to return HR images, there is no big visual change between different scale.

---

### Official Review · Reviewer_cxpS · 2023-10-30

**Soundness:** 3 good
**Presentation:** 3 good
**Contribution:** 2 fair
**Rating:** 5
**Confidence:** 4

**Summary:**

This paper proposes Diff-SR, a novel approach using pre-trained diffusion generative models (DGMs) for arbitrary-scale image super-resolution (ASSR) tasks. The key idea behind Diff-SR is to inject a specific amount of noise into the low-resolution (LR) image before feeding it into the DGM's backward diffusion process. By controlling the noise injection level, Diff-SR can adapt a single DGM to handle diverse ASSR tasks without requiring fine-tuning or distillation. Through theoretical analysis, the authors introduce the concept of Perceptual Recoverable Field (PRF) to determine the optimal noise level for different scales. Experiments on standard datasets demonstrate Diff-SR's effectiveness over competing methods.

**Strengths:**

- Interesting idea of utilizing noise injection to activate ASSR capacity in pre-trained DGMs, avoiding costly retraining or distillation.
- Great analysis of the impact of noise injection, leading to the valuable concept of PRF for quantifying model capacity.
- Superior performance over baselines on various metrics and datasets, especially for high upsampling rates.
- Qualitative results exhibit excellent recovery of textures and details.
- Great ablation studies validate key designs like adapting noise levels and DGM architectures.

**Weaknesses:**

- In my understanding, the claimed arbitrary-scale image super-resolution refers to the process that resizes the arbitrary LR input to 256*256  before restoring it. This claim may not be considered a novel approach. Many existing super-resolution methods can adopt a similar process, and this approach resembles the way blind restoration methods work.
- One notable drawback of this approach is its relatively high computational cost, primarily because it heavily relies on the iterative sampling process of diffusion generative models (DGMs) without employing approximations or early stopping techniques.
- The ablation studies provided in the paper seem to be narrowly focused on variations in model architecture, rather than rigorously examining different aspects of the methodology itself. A more comprehensive set of ablation studies that investigate various parameters of the approach could provide a more in-depth evaluation of its strengths and weaknesses, as these Diffusion based models usually have obviously larger parameters than this one (500 vs 35).
- Recent diffusion based super-resolution methods are missing. It is better to give analyses with them. The related works are not enough.

**Questions:**

- I note that the authors focus on image super-resolution tasks only. I wonder whether it can be used in real-world LR images, which usually contain different types of degradation, like noise, blur, and compression.
- Diff-SR contains 35.71 M parameters, which is obviously smaller than the competing diffusion based methods. In this task, do the model parameters have an obvious effect on the performance?
- The quantitative results in Table 1 seem to be not consistent with the results shown in the original papers. Do they adopt the same experiment settings?
- For LR images under the same downsampling scale, do they have the same parameters to balance the texture fidelity and semantic signature? If so, why? As the LR images which have the same downscale operation, may have different levels of blur. I think the trade-off parameters should also relate to the blur degree.

---

> ### Author Response · Authors · 2023-11-16
>
> Q1: Upsampling LR images to match the target pixel size is a commonly used technique in diffusion-based SR methods. While this technology is not new, our main contribution lies in combining upsampling with noise injection to utilize diffusion model for ASSR tasks. We have also conducted a mathematical analysis to explain why this approach can be effective. We acknowledge that our previous wording may have caused some confusion as Reviewer [c56f] said and we will be more careful with our choice of words and expressions in our revised version.
>
> | Model | Scale | DIV2K-FID | DIV2K-PSNR | DIV2K-SSIM |
> | ---- | ---- | ---- | ---- | ---- |
> | StableSR | 4x-downsample | 1.962 | 29.615 | 0.688 |
> | StableSR | 2x-downsample | 0.219 | 30.004 | 0.827 |
>
> Q2: The concern you raised regarding computational cost is a potential issue that exists in all current diffusion-based research. Our method naturally reduces some computational overhead by injecting noise only to a portion of the original image. Instead of starting from pure noise to generate the image, we begin the denoising process from the middle point, saving computational resources. Additionally, we have employed the DDIM sampler to reduce the total number of sampling steps to 50. Furthermore, consistency models suggest that one-step denoising can directly yield the original image from any intermediate result. Although we consider this as a potential solution for computational overhead, we leave it as future work.
>
> Q3: Our codebase is built upon the original DDPM [1]. In this codebase, the base-channel for the U-Net is set to 64. However, the latest DDPM codebases have larger base-channels, such as 256 (DDRM, DDNM) and 320 (LDM). This difference in parameter settings creates a gap between our model and other DGM models. Due to limitations in computing resources and time, it is currently challenging for us to train with a higher base-channel. We did attempt to reduce the base-channel from 64 to 32 to alleviate training overhead, but this led to a significant decrease in model performance.
>
> Q4: DDNM and DDRM are two recent studies that we consider relevant to our work. We acknowledge that StableSR is a novel diffusion-based SR model, and we have included its results for comparison. However, it is important to note that the experimental comparison may not be entirely fair, as StableSR can only upscale LR images to 512, while our experimental settings require a target pixel size of 256x256. Therefore, we had to downsample the StableSR image to match our target size. We provide this result as a reference, understanding that the comparison may not be entirely equitable.
>
> Q5: As your suggested, we followed the degradation pipeline designed by ESRGAN and tested it using the corresponding degradation code on DIV2K. We applied Blur, JPEG compression, and noise methods to degrade the original image. We then injected noise and denoised the image to evaluate the recovery effect of the model. Additionally, we used the two-order pipe from ESRGAN-pipe to degrade the image multiple times and finally reduced the image by 2 times. Interestingly, when using only a single type of degradation, the images were still repaired effectively. Among the mentioned degradations, the model performed best in recovering blur, but struggled the most with compression at 4x scale. Furthermore, we observed that the degradation results of image processing with a single degradation method were not visually obvious, especially in the case of JPEG compression. However, when applying ESRGAN's pipeline to degrade images multiple times, the model's performance significantly declined. Therefore, image recovery with multiple degradations remains a challenging task.
>
> |Method| FID | PSNR | SSIM |
> | ---- | ---- | ---- | ---- |
> |4x| 1.088 | 32.450 | 0.924 |
> |Blur| 0.331 | 33.583 | 0.953 |
> |JPEG Compression | 0.127 | 32.295 | 0.954 |
> |Noise| 0.104 | 32.220 | 0.947 |
> |ESRGAN| 1.528 | 30.817 | 0.821 |
>
> Q6: Yes, all of our experiments were conducted using the same experimental settings.
>
> Q7: In our experiments, we used the same compression method for the same compression ratio. Therefore, we assumed that all the images had the same blur level. We also used the same parameters to inject noise and denoise the images during the enhancement process. However, when entering the formula (19) that we derived, we realized that the decision of noise level is actually related to the difference between the high-resolution (HR) and low-resolution (LR) images. If the initial blur level of the image is different, the image with a higher degree of blur will experience a greater fidelity loss, requiring more noise injection. From this perspective, the degree of blur does play an important role in the trade-off. Unfortunately, we were unable to find suitable mathematical tools to establish a connection between the Fidelity term and the degree of blur in this paper.
>
> [1] https://github.com/lucidrains/denoising-diffusion-pytorch

---

> ### Comment · Reviewer_cxpS · 2023-11-20
>
> Thanks for the authors' response.
>
> Q1: The claim of arbitrary-scale super-resolution (ASSR) is strange, as mentioned by reviewer oyvV, as this work generates the results with a fixed resolution. Generally, the ASSR works can generate results with any resolution, or any scale, rather than the fixed resolution. If this can be regarded as ASSR, nearly all of the restoration works can be regarded as so. Please revise it carefully.
>
> Q2~Q3: It is well addressed.
>
> Q4: You can give some visual comparisons in the main paper by submitting a revised version.
>
> Q5: Please give the visual comparison in the main paper.
>
> Q6: I mean why the given quantitative results are not consistent with the original paper, as they have the same experiment settings?
>
> Q7: It is well addressed.

---

> ### Author Response · Authors · 2023-11-20
>
> For Q1, I will revise the claim according to your advice, but I am not sure whether I can revise the title.
>
> For Q4&Q5, I will add more visual comparisons in my revised version.
>
> For Q6: If you mean our arxiv version, the difference is because we use different datasets for evaluation since in our last version we are suggested to use more common datasets for evaluation.

---

> > ### Comment · Reviewer_cxpS · 2023-11-20
> >
> > For Q4 and Q5, I mean you can submit the revised version now. ICLR allows to submit the new version during the rebuttal period. You can add the visual comparison in the revised version.
> >
> > For Q6, I mean the quantitative result listed in Table 1 seems not consistent with the results in the competing methods. For example, the PSNR of SwinIR on benchmark Set5 of the X2 task is 38.42, which is better than 38.37 in this work. However, the result of SwinIR shown in this paper is 37.925. That's why I want to make sure whether you adopt the same settings for evaluation.

---

> > > ### Author Response · Authors · 2023-11-21
> > >
> > > Thanks for your reminder. We have uploaded the revise version. For the modifaction:
> > >
> > > First, we change the ASSR to TRSR (Target-resolution super-resolution) and the corresponding description over the whole paper. Meanwhile, we remove some exaggerated words in this revised version. However, we are not sure that whether we can change the title in the PDF during the rebuttle, so we leave it unchanged. We will revise the title after the decision.
> > >
> > > For Q4 and Q5, we add the visual comparison in the supplemetary file due to the limitation of paper size (9 paper) and file size (50 MB).
> > >
> > > For Q6, as we have state in the experiment, we do not use the original file in these datasets for comparison, we apply preliminary processing steps, including center cropping and resizing the images to a standardized size of $256\times256$. So the result may not be consistent with the original papers.

---

> > > > ### Comment · Reviewer_cxpS · 2023-11-21
> > > >
> > > > I really appreciate the authors' response. Honestly, the visual results of this work obviously inferior to StableSR. I understand they may not be entirely equitable. Let me consider the comments of other reviewers.

---

> > > > > ### Comment · Reviewer_cxpS · 2023-11-21
> > > > >
> > > > > By the way, I found that this paper is similar to DifFace, which also adjusts fidelity and perception through the number of steps (Noise variance). However, this is not discussed in this paper.
> > > > >
> > > > > DifFace: Blind Face Restoration with Diffused Error Contraction. arXiv preprint arXiv:2212.06512, 2022.

---

> > > > > > ### Author Response · Authors · 2023-11-21
> > > > > >
> > > > > > Thanks for your information, we have not found this paper before our submission.
> > > > > >
> > > > > > To my knowledge, this paper use a neural network to estimate the intermediate noise image with LR images as input, while we directly inject noise to the LR images. We all find that the intermediate noise images contain the capacity to recover HR images. However, this paper pay more attention on complex degradation settings which our focus is on the change of the SR scale. Besides, we also try to give some mathmatical proof to explain why this can work.
> > > > > >
> > > > > > We do thanks for your advice to help us improve our work.

---

> > > > > > > ### Comment · Reviewer_cxpS · 2023-11-21
> > > > > > >
> > > > > > > DifFace is available on Dec 2022. The two papers have very similar insights. Both inject noise on the LR image (or use CNN to predict an HR first) and adjust the noise level to achieve a fidelity-perception trade-off. DifFace also tries to explain it from a mathematical perspective about how can it work. Although most of my concerns are addressed, considering the inappropriate claim about ASSR and the similarity with DifFace but not discussed in the paper, I decided to keep the score unchanged.

---

### Official Review · Reviewer_c56f · 2023-11-01

**Soundness:** 2 fair
**Presentation:** 1 poor
**Contribution:** 2 fair
**Rating:** 5
**Confidence:** 3

**Summary:**

This paper provides an interesting way to use the pretrained diffusion model as tool to achieve arbitrary scale super-resolution. The idea of using a pretrained diffusion model to achieve such a task is novel to my knowledge. But the writing of the paper can take some work. The claimed theoretical contribution is not very convincing.

**Strengths:**

- The task of using pretrained diffusion model for arbitrary scale super resolution is very interesting.
- the core of the method is very simple : similar to SDEdit, where the low resolution image will be first upsampled and send forward through the diffusion pass before getting denoised. The diffusion model will try to add back the detail. If the noise is chosen properly, then the diffusion model can add detail without losing the structure.

**Weaknesses:**

- The writing of the paper can take some more work. There are many places that has the risk of overclaiming. For example, I don’t believe the methodology of this paper is “Pioneering” as it’s very similar to SDEdit and many other diffusion model techniques that add noise to the images before denoising it. I’m not very convinced of the mathematical “guarantee” as the foundation of the analysis seems to be built on equation (3), which is an inequality that can have slacks in between. Since I haven’t checked all the maths in detail I will defer to other reviewer. Some of the observations are already made by prior works (e.g. the message from Figure 1 and Figure 2(b) resembles the message in Figure 2 of the latent diffusion paper). I am not convinced that these observations are first discovered by this paper, so properly citing prior works could be appreciated.
- While it’s a good idea to leverage the pretrained diffusion model for downstream tasks, I believe the most powerful diffusion models are trained with text-conditioning and classfier-free guidance. I probably miss the proper discussion of how the method proposed by this method works with the state-of-the-art diffusion models of that kind.

**Questions:**

- “recovering LR images by generating visual details” do you mean recovering high-resolution images from LR images?

---

> ### Author Response · Authors · 2023-11-16
>
> Q1: Thank you for your suggestion. Regrading of the “pioneering”, our main intention is to convey that the use of pre-trained DGM to implement downstream ASSR without distillation and fine-tuning is a novel method that offers a fresh perspective on DGM usage. We apologize for any misunderstanding caused by the term "pioneering technology." In our revised version, we will be more careful with our choice of words and expressions.
>
> Q2: We agree with your idea that prior works should be properly cited. However, when designing Figure 1 and Figure 2, we did not refer to the design of Figure 2 in LDM. Figure 2(a) draws inspiration from the Figure 2 of consistency models [1], which we have cited in the references. Additionally, we have cited the LDM paper. Furthermore, we do not claim to be the first to discover this, as the Diffusion model has seen rapid development in the past two years, with numerous new papers being published daily. It is not surprising that many papers may have made similar observations. We have referenced all the papers we found during our research. However, due to the fast-paced nature of the latest research progress, it is possible that we may have missed some papers.
>
> Q3: We have also attempted to apply our ideas to LDM. However, we encountered a problem when transplanting this solution to LDM. Most text-conditioning in LDM is latent-space based, which means that before conducting the Diffusion process, the image needs to be encoded by the VAE model. However, the compression of the VAE model presents a challenge. The VAE model acts as an amplifier, exacerbating the gap between the HR image and the LR image in the latent space, resulting in narrower PRF windows. Therefore, achieving the same effect as the native Diffusion model when using LDM becomes difficult.
>
> Q4: Yes, as you said, we actually meant to express the idea of "restoring a low-resolution image to a high-resolution image." Thank you for pointing this mistake out.
>
> [1] Song Y, Dhariwal P, Chen M, et al. Consistency models[J]. 2023.

---

### Meta-Review · Area_Chair_joPj · 2023-12-06

**Metareview:**

This manuscript is currently positioned at the borderline for acceptance. A majority of the reviewers, three out of four, have evaluated it as marginally failing to meet the required threshold for acceptance.

Reviewer aha2 rated the manuscript slightly above the threshold. However, this assessment lacks sufficient support. Additionally, aha2 has raised critical concerns regarding the method's effectiveness in processing complex degradations, which are not adequately addressed in the manuscript or dicussion. Reviewer oyvV has queried the methodology, specifically the approach for determining the appropriate level of noise injection. The authors have presented some viable values, yet the criteria for systematically selecting an optimal noise level remain unclear.
Reviewer cxpS has decided to maintain a rating slightly below the acceptance level. This decision stems from issues with the paper's claims about ASSR and its similarities with some existing methodologies like DifFace.

**Justification For Why Not Higher Score:**

The manuscript, as it currently stands, does not sufficiently meet the acceptance criteria. It would benefit from a more rigorous methodological approach, especially in terms of noise level determination, and a clearer differentiation from existing works in the field.

**Justification For Why Not Lower Score:**

A more detailed meta-review will be provided.

---

### Decision · Program_Chairs · 2024-01-16

Reject